# Biofloc Technology in Fish Aquaculture: A Review

**DOI:** 10.3390/antiox12020398

**Published:** 2023-02-06

**Authors:** Young-Bin Yu, Jae-Ho Choi, Ju-Hyeong Lee, A-Hyun Jo, Kyung Mi Lee, Jun-Hwan Kim

**Affiliations:** 1Department of Aquatic Life Medicine, Pukyong National University, Busan 48513, Republic of Korea; 2Department of Aquatic Life and Medical Science, Sun Moon University, Asan-si 31460, Republic of Korea; 3Aquaculture Industry Division, West Sea Fisheries Research Institute, National Institute of Fisheries Science, Incheon 22383, Republic of Korea

**Keywords:** biofloc, growth performance, hematological parameters, antioxidant and immune responses, disease resistance

## Abstract

The application of biofloc to fish species has several advantages, including the enhancement of production by increasing growth performance and survival rate and the improvement of fish aquaculture physiological activity. There has been a recent increase in biofloc addition to fish culture, and this review examines changes this causes to the survival and growth rate of fish and its economic feasibility. Physiological activity and disease resistance of biofloc-fed fish is being extensively studied. The hematological parameters and antioxidant and immune responses of fish fed biofloc were reviewed in this study, as well as their disease resistance by testing them for major specific diseases. Standards for effectively applying biofloc to fish aquaculture are also suggested.

## 1. The Necessity of Biofloc Technology in Fish Aquaculture

The rapid growth of the fish farming industry has been caused by the pressure to build intensive aquaculture farms and therefore improve productivity [1,2]. Total global fish production is projected to reach 196 million tons (Mt) by 2025, and within this, aquaculture is estimated to exceed the production of capture fisheries [3]. However, there are several problems associated with this development, such as a decline in animal welfare standards due to breeding in overcrowded conditions and frequent disease occurrence due to poor environmental conditions and a reduction in the disease resistance of fish due to stress [4]. Antibiotics and chemical disinfectants are often used excessively for the prevention and treatment of diseases, but this can lead to drug resistance and the possible evolution of super bacteria due to increased bacterial resistance [5]. Thorough regulation and supervision are therefore needed to avoid these issues, but this is complicated by the fact that regulatory frameworks for the use of antibiotics in fish farming vary widely from country to country, and there are many countries where they are not implemented at all [6]. In fish aquaculture, various environmentally sustainable technologies are being used along with cutting-edge intelligent technologies to build a sustainable aquaculture industry ecosystem [7].

In fish aquaculture production, higher productivity is achieved with greater volumes of feed, resulting in an increase in waste production, which incurs environmental and economic costs. A total of 20–30% of the total nitrogen entering aquaculture ponds remains in the fish biomass, with the remainder becoming a water pollutant, producing high levels of toxic substances such as ammonia and nitrite [8]. Biofloc technology (BFT) is a more environmentally sustainable technology that uses beneficial microorganisms to absorb the ammonia and nitrite produced by feed waste, feces, and urine, which are naturally generated in the metabolic process of aquatic products [9]. This facilitates a self-nitrification process in aquaculture systems without the exchange of stock water and is achieved by stimulating the growth of beneficial microorganisms that can then be utilized as a feed source for aquaculture species and can absorb nitrogenous waste [10]. Innovative aquaculture systems using BFT have been applied to many fish farms due to increasing concern about environmental pollution. The BFT system is an eco-friendly closed system, which has many advantages including no water exchange, improved water quality and fish production, and less contamination by external factors [11]. The BFT system requires strong aeration and carbon sources such as sucrose, glucose, and molasses, and it helps to maintain the water quality by improving the activity of microorganisms and the removal of ammonia [5].

According to FAO statistics, aquaculture production accounted for 47.9% of total fish production in 2019, with common carps and tilapia species accounting for 34.9% and 3.5% of that amount, respectively. Diatin et al. [12] suggest that the majority (62%) of global fish aquaculture production will come from freshwater species such as carp, catfish, and tilapia. Most of these species are suitable for the application of biofloc technology (BFT), as they occupy a high proportion of fish farming and are farmed in ponds [13]. BFT has been successfully applied to intensive aquaculture fish species including common carp (*Cyprinus carpio*), Nile tilapia (*O. niloticus*), a polyculture of silver carp (*Hypophthalmichthys molitrix*), and bighead carp (*Aristichtys nobilis*) [14]. Efforts are being made to convert land-based systems with flowing-water culture to biofloc systems; however, this is still in the initial stage of research. Biofloc is widely known to improve fish-feed conversion rates and efficiency, liver condition, growth performance, digestive enzyme activity, and the immune competency of fish species, which overall improve fish growth [15]. In addition, biofloc improves biosecurity and feeding management control in fish farming.

## 2. Survival Rate and Growth Performance of Fish Raised in Biofloc

The useful microorganisms contained in biofloc activate the digestive enzymes of fish and increase feed efficiency, thereby improving growth performance. Biofloc also has a positive effect on survival rate by improving fish immunity [8]; the survival rate of fish species raised in biofloc in fish aquaculture is shown in Table 1. Khanjani et al. [16,17] reported that the survival rate of *O. niloticus* cultured in biofloc with simple carbon sources such as molasses, starch, barley flour, and corn was significantly improved when compared to the control group, with the starch-treated biofloc having the highest survival rate. The improvement in the survival rate of *O. niloticus* is likely due to the stress reduction induced by the improvement in the water environment and the addition of the essential amino acids, fatty acids, and nutritional compounds found in biofloc. Ekasari et al. [10] reported that the larval survival rate of *O. niloticus* broodstock cultured with biofloc was higher than that of the control, indicating that the BFT has a positive effect on larvae in Nile tilapia culture. Fauji et al. [18] reported that the survival rate of African catfish, *Clarias gariepinus*, raised in biofloc was 96% in the low-density section (4 fish/L), which was higher than that of the control (87%). However, there was a survival rate of 78% in the high-density (8 fish/L) biofloc, which is lower than that of the control group. These results highlight the importance of establishing an optimal density of fish culture using BFT and how vital it is to appreciate that if the density of effective microorganisms exceeds the water purification ability, there may be an adverse effect on the fish. Dauda et al. [19] found that the survival rates of *C. gariepinus* treated with glycerol and sucrose-treated biofloc were 90.6% and 76.3%, which were significantly higher than that of the control group (60.0%). However, the survival rate in the rice-bran-treated biofloc was found to be significantly lower (27%) than that of the control, which may have been due to the lack of carbon availability in the formation of biofloc. Therefore, it is important to use an appropriate carbon source that is suitable for farm environments and cultured organisms when using biofloc in fish culture. Haridas et al. [20] reported that biofloc significantly improved the survival rate of gray mullet, *Mugil cephalus*, particularly in the nursery phase. In other studies, it has been reported that both biofloc and control groups showed either 100% survival or no significant difference in the survival rate, which implies that biofloc has an effect on improving the immunity and health of fish but does not always show a significant difference.

The growth performance and feed conversion rate (FCR) of fish species raised in biofloc in fish aquaculture are demonstrated in Table 2. Azim and Little [21] reported that the growth of *O. niloticus* raised in biofloc increased by 44–46% compared to the control, which explains why biofloc is a suitable food source for fish. The growth performance of *O. niloticus* cultured with biofloc was increased at salinities of 4 and 8 g/L but decreased at salinities of 12 and 16 g/L, which indicates that growth when cultured with biofloc can vary greatly depending on salinity conditions [22]. Kishawy et al. [23] reported that the increase in the growth performance of *O. niloticus* cultured with biofloc using glycerol and Mannanoligosaccharides (MOS) as carbon sources was 11.72% and 27.57%, respectively, compared to the control, suggesting that the degree of growth improvement according to carbon source can vary. MOS is a prebiotic unable to be digested by fish enzymes but that can be digested by microbial enzymes. When complex carbohydrates (prebiotics) are added as a carbon source of biofloc, the nutritional content of the biofloc is more improved than that of glycerol. Luo et al. [24] reported that the growth rate of *O. niloticus* cultured with biofloc was 22% higher than that of *O. niloticus* cultured with recirculating aquaculture systems (RAS), suggesting that BFT could be a more effective method than RAS, another environmentally sustainable technique. Mirzakhani et al. [25] reported that the growth of *O. niloticus* cultured with biofloc was between 71.8% and 319.9% higher than that of the control and demonstrated the value of biofloc as a food source. Wang et al. [26] reported an increase in the growth of crucian carp, *Carassius auratus*, cultured with biofloc; higher increases in growth were observed as the C/N ratio increased. The results of this study confirmed that an appropriate increase in the C/N ratio stimulates the growth of biofloc, thereby increasing the growth of fish. Kim et al. [27] reported an increase in the growth of the olive flounder, *Paralichthys olivaceus,* of 26.3% when compared to controls, and they suggest that biofloc could improve the immunity and growth capacity of fish. Many authors reported a decrease in FCR with an increase in growth rate, suggesting that the biofloc environment increases the feed efficiency and feed conversion rate of fish. When *O. niloticus* was cultured with BFT and RAS, it was reported that the FCR of the biofloc was 1.20 ± 0.03, which was 18% higher than the FCR of 1.47 ± 0.02 of the RAS [24].

**Table 1 antioxidants-12-00398-t001:** Survival rate of fish species raised in biofloc in fish aquaculture.

Species	Carbon Source	C:N Ratio	Period	Change of Survival Rate	Reference
Freshwater	*Oreochromis niloticus*	Molasses	15:1	30 days	+	[16]
Starch	+
Barley flour	+
Corn	+
Molasses	8.4:1	12 weeks	×	[22]
Rice bran	15:1	10 weeks	×	[28]
Wheat-milling by-product	×
Sucrose	>10:1	87 days	×	[25]
Glucose	15:1	8 weeks	×	[29]
Wheat flour (200 fish/m^3^)	15:1	90 days	×	[30]
Wheat flour (250 fish/m^3^)	×
Wheat flour (300 fish/m^3^)	×
Wheat flour (350 fish/m^3^)	×
Molasses	10:1	14 days	+	[10]
100% molasses	15:1, 20:1	8 weeks	×	[26]
100% wheat flour	×
75% molasses + 25% wheat flour	×
50% molasses + 50% wheat flour	×
25% molasses + 75% wheat flour	×
Molasses	15:1	37 days	+	[17]
Starch	+
Barley flour	+
Corn	+
Molasses (40 fish/m^3^)	15:1	112 days	×	[31]
Molasses (80 fish/m^3^)	×
*Cyprinus carpio*	Sugar (6 kg/m^3^)	15:1	49 days	×	[32]
Sugar (12 kg/m^3^)	×
Rice bran	20:1	60 days	×	[33]
Sugarcane molasses	-
Rice bran + sugarcane molasses	×
Corn starch	15:1	60 days	×	[34]
*Cyprinus carpio* L.	Sugar beet molasses	20:1	70 days	×	[35]
Sugar	×
Corn starch	×
Molasses	20:1	30 days	×	[36]
*Clarias gariepinus*	Tapioca (4 fish/L)	10:1	20 days	+	[19]
Tapioca (6 fish/L)	-
Tapioca (8 fish/L)	-
Glycerol	15:1	8 weeks	×	[37]
Sucrose	15:1	6 weeks	+	[20]
Glycerol	+
Rice bran	-
*Carassius auratus*	Starch	15:1, 20:1	56 days	×	[27]
*Mugil cephalus*	Sucrose	15:1	60 days	+	[21]
*Heteropneustes fossilis*	Sugarcane molasses	10:1	120 days	×	[38]
Lemon fin barb hybrid (*Hypsibarbus wetmorei* 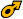 × *Barboides gonionotus* 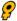 )	Glycerol	15:1	8 weeks	×	[37]
*Lepomis macrochirus*	Corn starch	15:1	32 days	-	[39]
Sucrose-sugar	-
*Labeo rohita*	Wheat flour	10:1	90 days	×	[40]
Seawater	*Oreochromis niloticus* sp.	Cornmeal + molasses (120 fish/m^3^)	15:1	7 weeks	×	[13]
Cornmeal + molasses (240 fish/m^3^)	×

+, increased, - decreased, and × no change in survival rate.

**Table 2 antioxidants-12-00398-t002:** Growth performance and FCR of fish species raised in biofloc in fish aquaculture.

Species	Carbon Source	C:N Ratio	Period	Response C:N Ratio	Response	Reference
Growth performance
Freshwater	*Oreochromis niloticus*	Molasses	8.4:1	12 weeks	8.4:1	+	[23]
Molasses (salinity level 4 g/L)	6:1	70 days	6:1	+	[23]
Molasses (salinity level 8 g/L)	6:1	+
Molasses (salinity level 12 g/L)	6:1	-
Molasses (salinity level 16 g/L)	6:1	-
Glycerol	15:1	12 weeks	15:1	+	[24]
Mannan oligosaccharides	15:1	+
Glucose (166 organisms/m^3^)	15:1	120 days	15:1	+	[41]
Glucose (333 organisms/m^3^)	15:1	+
Glucose (600 organisms/m^3^)	15:1	+
Glucose	10:1, 15:1, 20:1	120 days	10:1, 15:1	+	[42]
Rice bran and molasses (1:1) (60 fish/m^3^)	15:1	20 weeks	15:1	+	[43]
Rice bran and molasses (1:1) (80 fish/m^3^)	15:1	+
Molasses	15:1	30 days	15:1	+	[16]
Starch	15:1	+
Barley flour	15:1	+
Corn	15:1	+
Rice bran	15:1	10 weeks	15:1	+	[28]
Wheat-milling by-product	15:1	+
Wheat flour (200 fish/m^3^)	15:1	90 days	15:1	+	[30]
Wheat flour (250 fish/m^3^)	15:1	+
Wheat flour (300 fish/m^3^)	15:1	+
Wheat flour (350 fish/m^3^)	15:1	+
Sucrose	>10:1	87 days	> 10:1	+	[25]
Glucose	15:1	8 weeks	15:1	+	[29]
Molasses	10:1	14 days	-	×	[18]
100% molasses	15:1, 20:1	8 weeks	15:1, 20:1	+	[26]
100% wheat flour	15:1, 20:1	+
75% molasses + 25% wheat flour	15:1, 20:1	+
50% molasses + 50% wheat flour	15:1, 20:1	+
25% molasses + 75% wheat flour	15:1, 20:1	+
Molasses	15:1	37 days	15:1	+	[17]
Starch	15:1	+
Barley flour	15:1	+
Corn	15:1	+
Molasses (40 fish/m^3^)	15:1	112 days	15:1	+	[31]
Molasses (80 fish/m^3^)	15:1	+
*Cyprinus carpio*	Sugar (6 kg/m^3^)	15:1	49 days	15:1	+	[32]
Sugar (12 kg/m^3^)	15:1	+
Glucose	20:1	8 weeks	20:1	+	[8]
Corn starch	15:1	60 days	-	×	[34]
Rice bran (4.5 kg/m^3^)	20:1	60 days	20:1	+	[33]
Sugarcane molasses (4.5 kg/m^3^)	-	×
Rice bran + sugarcane molasses (4.5 kg/m^3^)	20:1	+
*Cyprinus carpio*	Sugar beet molasses	20:1	70 days	-	×	[35]
Sugar	-	×
Corn starch	-	×
Sugar beet molasses	20:1	10 weeks	-	×	[44]
Sugar	-	×
Corn starch	-	×
Molasses	20:1	30 days	20:1	+	[36]
*Clarias gariepinus*	Tapioca (4 fish/L)	10:1	20 days	10:1	+	[19]
Tapioca (6 fish/L)	10:1	+
Tapioca (8 fish/L)	10:1	+
Sucrose	15:1	6 weeks	-	×	[20]
Glycerol	-	×
Rice bran	-	×
Glycerol	15:1	8 weeks	-	×	[37]
*Carassius auratus*	Starch	15:1, 20:1, 25:1	56 days	20:1, 25:1	+	[27]
*Carassius auratus gibelio*	Glucose	20:1	8 weeks	20:1	+	[8]
*Mugil cephalus*	Sucrose	15:1	60 days	15:1	+	[21]
*Heteropneustes fossilis*	Sugarcane molasses	10:1	120 days	10:1	+	[29]
Lemon fin barb hybrid (*Hypsibarbus wetmorei* 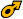 × *Barboides gonionotus* 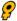 )	Glycerol	15:1	8 weeks	15:1	+	[37]
*Lepomis macrochirus*	Corn starch	15:1	32 days	15:1	-	[39]
Sucrose-sugar	15:1	-
*Labeo rohita*	Tapioca	15:1	60 days	15:1	+	[45]
Wheat	15:1	+
Corn	15:1	+
Sugar bagasse	15:1	+
Seawater	*Paralichthys olivaceus*	Glucose	<10:1	4 months	10:1	+	[27]
*Oreochromis niloticus* sp.	Cornmeal + molasses 120 fish/m^3^)	15:1	7 weeks	-	×	[13]
Cornmeal + molasses (240 fish/m^3^)	-	×
Feed conversion rate (FCR)
Freshwater	*Oreochromis niloticus*	Molasses	8.4:1	12 weeks	8.4:1	-	[22]
Molasses (salinity level 4 g/L)	6:1	70 days	-	×	[23]
Molasses (salinity level 8 g/L)	-	×
Molasses (salinity level 12 g/L)	-	×
Molasses (salinity level 16 g/L)	6:1	+
Glycerol	15:1	12 weeks	15:1	-	[24]
Mannan oligosaccharides	15:1	-
Glucose (166 organisms/m^3^)	15:1	120 days	15:1	-	[41]
Glucose (333 organisms/m^3^)	15:1	-
Glucose (600 organisms/m^3^)	15:1	-
Glucose	10:1, 15:1, 20:1	120 days	10:1, 15:1	+	[42]
Rice bran and molasses (1:1) (60 fish/m^3^)	15:1	20 weeks	15:1	-	[43]
Rice bran and molasses (1:1) (80 fish/m^3^)	15:1	-
Rice bran	15:1	10 weeks	15:1	-	[28]
Wheat-milling by-product	15:1	-
Sucrose	> 10:1	87 days	> 10:1	-	[25]
Glucose	15:1	8 weeks	15:1	-	[43]
100% molasses	15:1, 20:1	8 weeks	15:1, 20:1	-	[26]
100% wheat flour	15:1, 20:1	-
75% molasses + 25% wheat flour	15:1, 20:1	-
50% molasses + 50% wheat flour	15:1, 20:1	-
25% molasses + 75% wheat flour	15:1, 20:1	-
Molasses	15:1	37 days	15:1	+	[17]
Starch	15:1	+
Barley flour	15:1	+
Corn	15:1	+
Molasses (40 fish/m^3^)	15:1	112 days	15:1	+	[33]
Molasses (80 fish/m^3^)	15:1	+
*Cyprinus carpio*	Sugar (6 kg/m^3^)	15:1	49 days	15:1	-	[32]
Sugar (12 kg/m^3^)	15:1	-
Glucose	20:1	8 weeks	20:1	-	[8]
Corn starch	15:1	60 days	15:1	-	[34]
Rice bran	20:1	60 days	20:1	-	[33]
Sugarcane molasses	-	×
Rice bran + sugarcane molasses	20:1	-
Sugar beet molasses	20:1	70 days	20:1	-	[35]
Sugar	20:1	-
Corn starch	20:1	-
Sugar beet molasses	20:1	10 weeks	20:1	-	[44]
Sugar	20:1	-
Corn starch	20:1	-
*Clarias gariepinus*	Tapioca (4 fish/L)	10:1	20 days	10:1	-	[19]
Tapioca (6 fish/L)	10:1	-
Tapioca (8 fish/L)	10:1	-
Glycerol	15:1	8 weeks	-	×	[37]
*Carassius auratus*	Starch	15:1, 20:1, 25:1	56 days	20:1, 25:1	-	[27]
*Carassius auratus gibelio*	Glucose	20:1	8 weeks	20:1	-	[8]
*Mugil cephalus*	Sucrose	15:1	60 days	15:1	-	[21]
*Heteropneustes fossilis*	Sugarcane molasses	10:1	120 days	10:1	-	[38]
Lemon fin barb hybrid (*Hypsibarbus wetmorei* 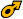 × *Barboides gonionotus* 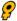 )	Glycerol	15:1	8 weeks	15:1	-	[37]
*Labeo rohita*	Tapioca	15:1	60 days	15:1	-	[45]
Wheat	15:1	-
Corn	15:1	-
Sugar bagasse	15:1	-
Seawater	*Oreochromis niloticus* sp.	Cornmeal + molasses (120 fish/m^3^)	15:1	7 weeks	-	×	[13]
Cornmeal + molasses (240 fish/m^3^)	-	×

+, increased, - decreased, and × no change in survival rate.

## 3. Hematological Parameters

Blood in the circulatory system serves several functions in the survival of fish, and hematological parameters are essential indicators for evaluating their physiological state, stress, immune responses, and disease resistance, as well as reflecting the nutritional and environmental conditions, thereby detecting abnormalities connected to fish health status [38]. Biofloc can contribute to the improvement of the health of fish due to both its digestive enzyme activity and physiological activity, and the evaluation of hematological parameters should be a good indicator to confirm its efficacy [18]. Hematological and biochemical parameters of fish species raised in biofloc are demonstrated in Table 3. Erythrocytes are the most abundant cellular components in the circulatory system of fish, playing vital roles in gas exchange and the respiration of cells, as well as performing several functions related to immunity, such as antiviral responses, phagocytosis or cytokine-mediated signaling [46]. Red blood cells (RBCs) can synthesize proteins such as hemoglobin, and an increase in them indicates good health, protecting the fish from stress and disease conditions via non-specific immune responses [38]. Shourbela et al. [47] reported that biofloc made from various carbon sources such as glycerol, molasses, and starch induced a significant increase in RBCs in *O. niloticus*. Fauji et al. [19] also reported a significant increase in RBCs in *C. gariepinus* cultured with biofloc, and they go on to suggest that it was therefore healthier than the control group. However, in most studies, there was no significant change in the number of RBCs due to the addition of biofloc to the environment, which suggests that it has little effect on the physiology of fish [29,31,34,35].

While RBCs are responsible for gas exchange, leukocytes (white blood cells: WBCs) are circulating cells of the immune system that are involved in both innate and acquired immune responses by expressing cell-specific immune-related genes [48]. WBCs are associated with the regulation of immune functions, and their number may increase as a protective mechanism during the stress response [49]. Lymphocytes are the most common type of WBCs, and the interaction between lymphocytes B and T is required for an immune response to occur, meaning that the increase in white blood cells is an effective indicator of the stimulation of the fish immune system [50]. Mansour and Esteban [28] reported a significant increase in WBCs in *O. niloticus* cultured in biofloc, which was caused by an increase in leukocytes due to there being more neutrophils and lymphocytes. This increase was induced by the higher protein levels in the biofloc environment. Other authors have also reported significant increases in WBCs in *C. gariepinus* and Asian stinging catfish, *Heteropneustes fossilis*, when cultured in biofloc, suggesting that it causes a higher immune capacity and better health in fish [19,38]. However, many studies showed either no change or a decrease in many fish raised with biofloc, and some suggest that there was no effect on health.

Hematocrit and hemoglobin are both important indicators for evaluating fish health [51,52]. Mansour and Esteban [28] reported a significant increase in hematocrit in *O. niloticus* cultured with biofloc made from carbon sources of rice bran and wheat-milling by-product, but there was no significant change in biofloc made with glucose as a carbon source. Most of the studies however did not show changes in hematocrit, indicating that the biofloc system did not adversely affect the hematological properties of fish. The main function of hemoglobin is to transport oxygen from the gas exchange organ to peripheral tissues, and a decrease in hemoglobin under stress conditions can be manifested by a decrease in the rate of hemoglobin synthesis, leading to impaired oxygen supply to the tissues and eventually resulting in a decrease in RBCs via hemolysis [53,54]. Shourbela et al. [48] reported a significant increase in the hemoglobin of *O. niloticus* cultured with biofloc made from various carbon sources such as glycerol, molasses, and starch, indicating an improved health status of fish in this culture. Fauji et al. [19] also reported increased hemoglobin levels in *C. gariepinus* cultured with biofloc, suggesting that biofloc has a physiologically positive effect on fish [31,38,55].

Blood glucose levels are used by biological systems as an essential fuel to enhance muscle activity, and glucose is an important indicator in assessing acute stress, as glucose levels increase when tissues such as the brain, gills, and muscles increase respiration to cope with the increased energy demands of stress [56]. In a stressful environment, stress hormones such as cortisol and catecholamine are released as a primary reaction in fish, with glucose production being a secondary reaction [57]. Shourbela et al. [47] showed that the plasma glucose of *O. niloticus* cultured with biofloc made from various carbon sources such as glycerol, molasses, and starch was significantly decreased, meaning that stress levels were lower in the fish in biofloc compared to the control group. Verma et al. [45] also discovered a decrease in plasma glucose levels in Rohu, *Labeo rohita,* when it was cultured with biofloc made from many carbon sources such as tapioca, wheat, corn, and sugar bagasse, and they suggest that this was due to a decrease in cortisol and glucose caused by the less stressful biofloc environment. Sontakke et al. [58] also reported a significant decrease in plasma glucose in milkfish, *Chanos chanos*, cultured with biofloc made of various carbon sources such as sorghum, potato, yam, and glucose, and Kim et al. [27] showed a significant decrease in plasma glucose in olive flounder, *Paralichthys olivaceus*, cultured with biofloc, similarly suggesting that this decrease means that the fish raised with biofloc had less physiological stress than controls. Most studies have reported low plasma glucose levels in fish raised with biofloc, but some studies have reported an increase, suggesting that the biofloc environment may act as a stressor depending on the species and conditions.

Cholesterol is a major component of cell membranes as well as a precursor of all steroid hormones, and it is a major indicator of the health status of fish [59]. Although some studies showed a significant increase or decrease in fish cultured with biofloc, most studies did not show changes in the plasma cholesterol in fish, suggesting that there is no adverse physiological effect [27,33,34,35,60]. Plasma total proteins, including albumin and globulins, are major compounds synthesized in the liver that play an important role in the immune response, meaning that an increase in plasma protein levels is associated with a stronger innate immune response in fish [61,62]. Mansour and Esteban [28] reported a significant increase in the plasma total protein of *O. niloticus* cultured with biofloc, which implies an improvement in the innate immune response. Verma et al. [45] reported a significant increase in *L. rohita* plasma glucose in biofloc made from tapioca, but a significant decrease in plasma glucose in biofloc made from wheat, corn, and sugar bagasse, meaning that fish raised in biofloc made from tapioca had a lower immune status compared to when it was made with wheat, corn, and sugar bagasse. Many studies have reported a significant increase in plasma total protein with biofloc in various fish species such as *C. carpio*, sutchi catfish, *Pangasianodon hypophthalmus,* and *C. chanos* [32,33,35,55,58]. However, in some studies, there is no change or a decrease in plasma glucose levels in fish cultured with biofloc, indicating that its efficacy may be limited.

Albumin and globulin are major proteins in the serum. Albumin is a protein carrier involved in the transport of various substances including lipids, hormones, and inorganic ions [63]. Globulin comprises a1, a2, β, and γ-globulin fractions and is a critical component for maintaining a healthy immune system in fish, as an increase in globulin levels is associated with a stronger innate immune response [64]. The ratio of albumin to globulin is a useful indicator for monitoring fish health and immune status. Mansour and Esteban [28] reported a significant increase in plasma albumin in *O. niloticus* cultured with biofloc. Many authors have reported a similar effect, but also that the change depends on the conditions, implying that there must be an appropriate carbon source and C/N ratio for it to be an immunostimulant [13,26,32,45,47]. Nageswari et al. [55] reported a significant increase in serum albumin in *P. hypophthalmus* cultured with biofloc made from various carbon sources such as tapioca, sorghum, pearl millet, and finger millet, potentially due to the increase in immunity caused by the bioactive compounds of biofloc. Sontakke et al. [58] also reported a significant increase in serum albumin of *C. chanos* cultured with biofloc made from various carbon sources such as sorghum, potato, yam, and glucose, and they suggest that this was due to the improved immunity of fish. Many authors have reported significant increases in plasma/serum globulin in various fish species such as *O. niloticus*, *L. rohita*, *C. carpio*, *P. hypophthalmus,* and *C. chanos* cultured with biofloc, and this increase means an improved immune status due to interactions with the immune stimulating agent (physiologically active substance) and effective microorganisms present in the biofloc [13,24,26,28,33,45,47,58].

Triglycerides, a provider of cellular energy, are major components of lipoproteins along with cholesterol and phospholipids, and they can be critical biomarkers in evaluating the nutritional status of fish metabolism [65]. Although some studies reported a significant increase in serum/plasma triglyceride in fish cultured with biofloc [20,32], most studies did not show a significant difference in triglyceride levels [33,35].

Aspartate aminotransferase (AST) and alanine aminotransferase (ALT) are responsible for the catalysis of interconversion of non-essential amino acids including glutamate, aspartate, and alanine in fish [66]. Since fish plasma AST and ALT are released into the blood by the increased permeability of damaged hepatocytes due to various environmental stresses and disease infections, the levels of these are important factors in diagnosing liver function and damage [47,67]. Adineh et al. [32] reported that the serum AST and ALT of *C. carpio* cultured with biofloc were significantly decreased, indicating lower breeding stress. Yu et al. [68] also reported significant reductions in plasma AST and ALT of Kaoping freshwater minnow, *Opsariichthys kaopingensis*. These suggest that the reduction in AST and ALT of *P. olivaceus* raised in biofloc creates less stress in the biofloc environment. Alkaline phosphatase (ALP), in addition to the pinocytic vesicle and Golgi complex, is a membrane-bound enzyme found in the bile pole of hepatocytes, which is an important enzyme in fish metabolism that transports metabolites across the membrane [69]. ALP activity plays a role in immune regulation and defense mechanisms in fish, and it is widely used as an indicator of stress-induced tissue damage and physiological responses [70]. Adineh et al. [32] reported a significant increase in serum ALP of *C. carpio* cultured with biofloc and suggested that the increase in enzymatic activity was due to the stimulated immune activity induced by the biofloc environment. During a stress response, cortisol secretion from interrenal cells in the head kidney is activated by corticotropin-releasing factors via the secretion of adrenocorticotropic hormone from the anterior pituitary, and this is a major indicator of stress in fish [71]. This cortisol increase leads to phagocytic and complement activity suppression in blood and head-kidneys, a decrease in the number of lymphocytes, and an increase in susceptibility to infection [72]. Most of the studies showed a significant decrease in serum cortisol in fish cultured with biofloc, and it was argued that this decrease was the result of proving that biofloc had an effect of relieving stress in fish [30,32,34,47,55,58]. Verma et al. [45] also reported a significant decrease in serum cortisol in *L. rohita* cultured with biofloc, suggesting that biofloc had an anti-stress effect on fish.

**Table 3 antioxidants-12-00398-t003:** Hematological and biochemical parameters of fish species raised in the biofloc in fish aquaculture.

Species		Carbon Source	C:N Ratio	Period	Target Organ	Response C/N Ratio	Response	Reference
Red blood cell (RBC)								
Freshwater	*Oreochromis niloticus*	Glycerol (140 fish/m^3^)	15:1	98 days	Blood	15:1	+	[47]
		Glycerol (280 fish/m^3^)				15:1	+	
		Molasses (140 fish/m^3^)	15:1	98 days	Blood	15:1	+	
		Molasses (280 fish/m^3^)				15:1	+	
		Starch (140 fish/m^3^)	15:1	98 days	Blood	15:1	+	
		Starch (280 fish/m^3^)				15:1	+	
		Glucose	15:1	8 weeks	Blood	-	×	[29]
		Molasses (40 fish/m^3^)	15:1	112 days	Blood	-	×	[31]
		Molasses (80 fish/m^3^)				-	×	
	*Cyprinus carpio*	Rice bran	20:1	60 days	Blood	-	×	[33]
		Sugarcane molasses				-	×	
		Rice bran + sugarcane molasses				20:1	+	
		Corn starch	15:1	60 days	Blood	-	×	[34]
		Corn starch (10% of daily feed deducted)				-	×	
		Sugar beet molasses	20:1	10 weeks	Blood	-	×	[44]
		Sugar				-	×	
		corn starch				-	×	
	*Clarias gariepinus*	Tapioca (4 fish/L)	10:1	20 days	Blood	-	×	[19]
		Tapioca (6 fish/L)				10:1	+	
		Tapioca (8 fish/L)				-	×	
	*Heteropneustes fossilis*	Sugarcane molasses	10:1	120 days	Blood	10:1	+	[38]
White blood cell (WBC)								
Freshwater	*Oreochromis niloticus*	Rice bran	15:1	10 weeks	Blood	15:1	+	[28]
		Wheat-milling by-product				15:1	+	
		Glycerol (140 fish/m^3^)	15:1	98 days	Blood	15:1	-	[47]
		Glycerol (280 fish/m^3^)				15:1	-	
		Molasses (140 fish/m^3^)	15:1	98 days	Blood	15:1	-	
		Molasses (280 fish/m^3^)				15:1	-	
		Starch (140 fish/m^3^)	15:1	98 days	Blood	15:1	-	
		Starch (280 fish/m^3^)				-	×	
		Glucose	15:1	8 weeks	Blood	-	×	[29]
	*Cyprinus carpio*	Rice bran	20:1	60 days	Blood	-	×	[33]
		Sugarcane molasses				-	×	
		Rice bran + sugarcane molasses				-	×	
		Corn starch	15:1	60 days	Blood	-	×	[34]
		Sugar beet molasses	20:1	10 weeks	Blood	-	×	[44]
		Sugar				-	×	
		corn starch				-	×	
	*Clarias gariepinus*	Tapioca (4 fish/L)	10:1	20 days	Blood	10:1	+	[19]
		Tapioca (6 fish/L)				10:1	+	
		Tapioca (8 fish/L)				10:1	+	
	*Heteropneustes fossilis*	Sugarcane molasses	10:1	120 days	Blood	10:1	+	[38]
Hematocrit (Ht)								
Freshwater	*Oreochromis niloticus*	Rice bran	15:1	10 weeks	Blood	15:1	+	[28]
		Wheat-milling by-product				15:1	+	
		Glucose	15:1	8 weeks	Blood	-	×	[29]
	*Cyprinus carpio*	Sugar beet molasses	20:1	10 weeks	Blood	-	×	[44]
		Sugar				-	×	
		Corn starch				-	×	
		Corn starch	15:1	60 days	Blood	-	×	[34]
	*Clarias gariepinus*	Tapioca (4 fish/L)	10:1	20 days	Blood	-	×	[19]
		Tapioca (6 fish/L)				-	×	
		Tapioca (8 fish/L)				-	×	
Seawater	*Paralichthys olivaceus*	Glucose	<10:1	4 months	Blood	-	×	[27]
Hemoglobin (Hb)								
Freshwater	*Oreochromis niloticus*	Glycerol (140 fish/m^3^)	15:1	98 days	Blood	15:1	+	[47]
		Glycerol (280 fish/m^3^)				15:1	+	
		Molasses (140 fish/m^3^)	15:1	98 days	Blood	15:1	+	
		Molasses (280 fish/m^3^)				15:1	+	
		Starch (140 fish/m^3^)	15:1	98 days	Blood	15:1	+	
		Starch (280 fish/m^3^)				15:1	+	
		Glucose	15:1	8 weeks	Blood	-	×	[29]
		Molasses (40 fish/m^3^)	15:1	112 days	Blood	15:1	+	[31]
		Molasses (80 fish/m^3^)				-	×	
	*Cyprinus carpio*	Rice bran (4.5 kg/m^3^)	20:1	60 days	Blood	-	×	[33]
		Sugarcane molasses (4.5 kg/m^3^)				20:1	-	
		Rice bran + sugarcane molasses (4.5 kg/m^3^)				-	×	
	*Clarias gariepinus*	Tapioca (4 fish/L)	10:1	20 days	Blood	10:1	+	[19]
		Tapioca (6 fish/L)				-	×	
		Tapioca (8 fish/L)				10:1	+	
	*Heteropneustes fossilis*	Sugarcane molasses	10:1	120 days	Blood	10:1	+	[38]
	*Pangasianodon hypophthalmus*	Tapioca	15:1	90 days	Blood	15:1	+	[54]
		Sorghum				15:1	+	
		Pearl millet				15:1	+	
		Finger millet				15:1	+	
Seawater	*Paralichthys olivaceus*	Glucose	<10:1	4 months	Blood	-	×	[27]
Glucose								
*Freshwater*	*Oreochromis niloticus*	Glycerol (140 fish/m^3^)	15:1	98 days	Serum	15:1	-	[48]
		Glycerol (280 fish/m^3^)				15:1	-	
		Molasses (140 fish/m^3^)	15:1	98 days	Serum	15:1	-	
		Molasses (280 fish/m^3^)				15:1	-	
		Starch (140 fish/m^3^)	15:1	98 days	Serum	15:1	-	
		Starch (280 fish/m^3^)				15:1	-	
		Sucrose	>10:1	87 days	Serum	-	×	[25]
		Wheat flour (200 fish/m^3^)	15:1	90 days	Serum	15:1	-	[30]
		Wheat flour (250 fish/m^3^)				15:1	-	
		Wheat flour (300 fish/m^3^)				15:1	-	
		Wheat flour (350 fish/m^3^)				15:1	-	
		Molasses (40 fish/m^3^)	15:1	112 days	Plasma	15:1	-	[31]
		Molasses (80 fish/m^3^)				-	×	
	Genetically Improved Farmed Tilapia	Spentwash	10:1	180 days	Serum	10:1	-	[73]
	*Labeo rohita*	Tapioca	15:1	20 days	Serum	15:1	-	[45]
				40 days		15:1	-	
				60 days		15:1	-	
		Wheat	15:1	20 days	Serum	15:1	-	
				40 days		15:1	-	
				60 days		15:1	-	
		Corn	15:1	20 days	Serum	15:1	-	
				40 days		15:1	-	
				60 days		-	×	
		Sugar bagasse	15:1	20 days	Serum	15:1	-	
				40 days		15:1	-	
				60 days		15:1	-	
		Molasses	15:1	16 weeks	Serum	-	×	[74]
	*Cyprinus carpio*	Sugar (6 kg/m^3^)	15:1	49 days	Serum	15:1	-	[32]
		Sugar (12 kg/m^3^)				15:1	-	
		Rice bran (4.5 kg/m^3^)	20:1	60 days	Serum	-	×	[33]
		Sugarcane molasses (4.5 kg/m^3^)				20:1	+	
		Rice bran + Sugarcane molasses (4.5 kg/m^3^)				-	×	
		Sugar beet molasses	20:1	10 weeks	Serum	-	×	[44]
		Sugar				-	×	
		Corn starch				-	×	
	*Clarias gariepinus*	Sucrose	15:1	6 weeks	Plasma	-	×	[20]
		Glycerol				-	×	
		Rice bran				-	×	
	*Pangasianodon hypophthalmus*	Tapioca	15:1	90 days	Serum	15:1	-	[55]
		Sorghum				15:1	-	
		Pearl millet				15:1	-	
		Finger millet				15:1	-	
Brackish water	*Mugil cephalus*	Sucrose	15:1	60 days	Serum	-	×	[30]
	*Chanos chanos*	Sorghum	15:1	45 days	Serum	15:1	-	[58]
				90 days		15:1	-	
		Potato	15:1	45 days	Serum	15:1	-	
				90 days		15:1	-	
		Yam	15:1	45 days	Serum	15:1	-	
				90 days		15:1	-	
		Glucose	15:1	45 days	Serum	15:1	-	
				90 days		15:1	-	
Seawater	*Paralichthys olivaceus*	Glucose	<10:1	4 months	Plasma	<10:1	-	[27]
	*Oreochromis* sp.	Cornmeal + molasses (120 fish/m^3^)	15:1	7 weeks	Plasma	-	×	[13]
		Cornmeal + molasses (240 fish/m^3^)				-	×	
Cholesterol								
Freshwater	*Oreochromis niloticus*	Molasses (40 fish/m^3^)	15:1	112 days	Plasma	-	×	[31]
		Molasses (80 fish/m^3^)				15:1	-	
	*Cyprinus carpio*	Sugar (6 kg/m^3^)	15:1	49 days	Serum	-	×	[32]
		Sugar (12 kg/m^3^)				15:1	+	
		Rice bran (4.5 kg/m^3^)	20:1	60 days	Serum	-	×	[33]
		Sugarcane molasses (4.5 kg/m^3^)				-	×	
		Rice bran + sugarcane molasses (4.5 kg/m^3^)				-	×	
		Corn starch	15:1	60 days	Plasma	-	×	[34]
		Sugar beet molasses	20:1	10 weeks	Serum	-	×	[44]
		Sugar				-	×	
		Corn starch				-	×	
Seawater	*Paralichthys olivaceus*	Glucose	<10:1	4 months	Plasma	-	×	[27]
	*Oreochromis* sp.	Cornmeal + molasses (120 fish/m^3^)	15:1	7 weeks	Plasma	15:1	-	[13]
		Cornmeal + molasses (240 fish/m^3^)				-	×	
Total protein								
Freshwater	*Oreochromis niloticus*	Rice bran	15:1	10 weeks	Plasma	15:1	+	[28]
		Wheat-milling by-product				15:1	+	
		Glycerol (140 fish/m^3^)	15:1	98 days	Serum	15:1	+	[47]
		Glycerol (280 fish/m^3^)				-	×	
		Molasses (140 fish/m^3^)	15:1	98 days	Serum	15:1	+	
		Molasses (280 fish/m^3^)				15:1	+	
		Starch (140 fish/m^3^)	15:1	98 days	Serum	-	×	
		Starch (280 fish/m^3^)				15:1	+	
		Glucose	15:1	8 weeks	Serum	-	×	[29]
		Mannan oligosaccharides	15:1	12 weeks	Serum	15:1	+	[24]
		Glycerol				15:1	+	
		100% molasses	15:1, 20:1	8 weeks	Serum	15:1, 20:1	+	[26]
		100% wheat flour				15:1, 20:1	+	
		75% molasses + 25% wheat flour				15:1, 20:1	+	
		50% molasses + 50 wheat flour				15:1, 20:1	+	
		25% molasses + 75% wheat flour				15:1, 20:1	+	
		Molasses (40 fish/m^3^)	15:1	112 days	Plasma	-	×	[31]
		Molasses (80 fish/m^3^)				15:1	-	
	Genetically Improved Farmed Tilapia	Spentwash	10:1	180 days	Serum	10:1	+	[73]
	*Labeo rohita*	Tapioca	15:1	20 days	Serum	-	×	[45]
				40 days		15:1	+	
				60 days		15:1	+	
		Wheat	15:1	20 days	Serum	15:1	-	
				40 days		-	×	
				60 days		-	×	
		Corn	15:1	20 days	Serum	15:1	-	
				40 days		-	×	
				60 days		15:1	-	
		Sugar bagasse	15:1	20 days	Serum	15:1	-	
				40 days		-	×	
				60 days		-	×	
		Molasses	15:1	16 weeks	Serum	-	×	[74]
	*Cyprinus carpio*	Sugar (6 kg/m^3^)	15:1	49 days	Serum	-	×	[32]
		Sugar (12 kg/m^3^)				15:1	+	
		Rice bran (4.5 kg/m^3^)	20:1	60 days	Serum	-	×	[33]
		Sugarcane molasses (4.5 kg/m^3^)				20:1	+	
		Rice bran + sugarcane molasses (4.5 kg/m^3^)				20:1	+	
		Corn starch	15:1	60 days	Plasma	-	×	[34]
		Sugar beet molasses	20:1	10 weeks	Serum	20:1	+	[44]
		Sugar				20:1	+	
		Corn starch				20:1	+	
	*Pangasianodon hypophthalmus*	Tapioca	15:1	90 days	Serum	15:1	+	[55]
		Sorghum				15:1	+	
		Pearl millet				15:1	+	
		Finger millet				15:1	+	
Brackish water	*Mugil cephalus*	Sucrose	15:1	60 days	Serum	-	×	[21]
	*Chanos chanos*	Sorghum	15:1	45 days	Serum	15:1	+	[58]
				90 days		15:1	+	
		Potato	15:1	45 days	Serum	-	×	
				90 days		15:1	+	
		Yam	15:1	45 days	Serum	15:1	+	
				90 days		15:1	+	
		Glucose	15:1	45 days	Serum	15:1	+	
				90 days		15:1	+	
Seawater	*Paralichthys olivaceus*	Glucose	<10:1	4 months	Plasma	-	×	[28]
	*Oreochromis* sp.	Cornmeal + molasses (120 fish/m^3^)	15:1	7 weeks	Plasma	15:1	-	[13]
		Cornmeal + molasses (240 fish/m^3^)				-	×	
Albumin								
Freshwater	*Oreochromis niloticus*	Rice bran	15:1	10 weeks	Plasma	15:1	+	[28]
		Wheat-milling by-product				15:1	+	
		Glycerol (140 fish/m^3^)	15:1	98 days	Serum	-	×	[47]
		Glycerol (280 fish/m^3^)				-	×	
		Molasses (140 fish/m^3^)	15:1	98 days	Serum	-	×	
		Molasses (280 fish/m^3^)				-	×	
		Starch (140 fish/m^3^)	15:1	98 days	Serum	-	×	
		Starch (280 fish/m^3^)				15:1	+	
		Mannan oligosaccharides	15:1	12 weeks	Serum	15:1	+	[24]
		Glycerol				15:1	+	
		100% molasses	15:1, 20:1	8 weeks	Serum	-	×	[26]
		100% wheat flour				15:1, 20:1	+	
		75% molasses + 25% wheat flour				-	×	
		50% molasses + 50 wheat flour				-	×	
		25% molasses + 75% wheat flour				-	×	
	*Labeo rohita*	Tapioca	15:1	20 days	Serum	15:1	-	[45]
				40 days		15:1	+	
				60 days		-	×	
		Wheat	15:1	20 days	Serum	15:1	-	
				40 days		-	×	
				60 days		15:1	-	
		Corn	15:1	20 days	Serum	15:1	-	
				40 days		15:1	+	
				60 days		15:1	+	
		Sugar bagasse	15:1	20 days	Serum	15:1	-	
				40 days		-	×	
				60 days		-	×	
	*Cyprinus carpio*	Sugar (6 kg/m^3^)	15:1	49 days	Serum	15:1	-	[32]
		Sugar (12 kg/m^3^)				15:1	+	
		Rice bran (4.5 kg/m^3^)	20:1	60 days	Serum	-	×	[33]
		Sugarcane molasses (4.5 kg/m^3^)				-	×	
		Rice bran + sugarcane molasses (4.5 kg/m^3^)				-	×	
		Corn starch	15:1	60 days	Plasma	-	×	[34]
		Sugar beet molasses	20:1	10 weeks	Serum	-	×	[44]
		Sugar				-	×	
		Corn starch				-	×	
	*Pangasianodon hypophthalmus*	Tapioca	15:1	90 days	Serum	15:1	+	[55]
		Sorghum				15:1	+	
		Pearl millet				15:1	+	
		Finger millet				15:1	+	
Brackish water	*Mugil cephalus*	Sucrose	15:1	60 days	Serum	-	×	[21]
	*Chanos chanos*	Sorghum	15:1	45 days	Serum	15:1	+	[58]
				90 days		15:1	+	
		Potato	15:1	45 days	Serum	15:1	+	
				90 days		15:1	+	
		Yam	15:1	45 days	Serum	15:1	+	
				90 days		15:1	+	
		Glucose	15:1	45 days	Serum	15:1	+	
				90 days		15:1	+	
Sea water	*Oreochromis* sp.	Cornmeal + molasses (120 fish/m^3^)	15:1	7 weeks	Plasma	15:1	-	[8]
		Cornmeal + molasses (240 fish/m^3^)				15:1	+	
Globulin								
Freshwater	*Oreochromis niloticus*	Rice bran	15:1	10 weeks	Plasma	-	×	[28]
		Wheat-milling by-product				15:1	+	
		Glycerol (140 fish/m^3^)	15:1	98 days	Serum	15:1	+	[47]
		Glycerol (280 fish/m^3^)				-	×	
		Molasses (140 fish/m^3^)	15:1	98 days	Serum	15:1	+	
		Molasses (280 fish/m^3^)				15:1	+	
		Starch (140 fish/m^3^)	15:1	98 days	Serum	-	×	
		Starch (280 fish/m^3^)				15:1	+	
		Mannan oligosaccharides	15:1	12 weeks	Serum	15:1	+	[24]
		Glycerol				15:1	+	
		100% molasses	15:1, 20:1	8 weeks	Serum	-	×	[26]
		100% wheat flour				15:1, 20:1	+	
		75% molasses + 25% wheat flour				-	×	
		50% molasses + 50% wheat flour				15:1	+	
		25% molasses + 75% wheat flour				15:1, 20:1	+	
	*Labeo rohita*	Tapioca	15:1	20 days	Serum	15:1	+	[45]
				40 days		-	×	
				60 days		15:1	+	
		Wheat	15:1	20 days	Serum	15:1	+	
				40 days		-	×	
				60 days		-	×	
		Corn	15:1	20 days	Serum	-	×	
				40 days		15:1	-	
				60 days		-	×	
		Sugar bagasse	15:1	20 days	Serum	-	×	
				40 days		-	×	
				60 days		-	×	
	*Cyprinus carpio*	Sugar (6 kg/m^3^)	15:1	49 days	Serum	-	×	[32]
		Sugar (12 kg/m^3^)				15:1	+	
		Rice bran (4.5 kg/m^3^)	20:1	60 days	Serum	20:1	+	[33]
		Sugarcane molasses (4.5 kg/m^3^)				20:1	+	
		Rice bran + sugarcane molasses (4.5 kg/m^3^)				20:1	+	
	*Pangasianodon hypophthalmus*	Tapioca	15:1	90 days	Serum	15:1	+	[55]
		Sorghum				15:1	+	
		Pearl millet				15:1	+	
		Finger millet				15:1	+	
Brackish water	*Chanos chanos*	Sorghum	15:1	45 days	Serum	15:1	+	[58]
				90 days		15:1	+	
		Potato	15:1	45 days	Serum	-	×	
				90 days		15:1	+	
		Yam	15:1	45 days	Serum	15:1	+	
				90 days		15:1	+	
		Glucose	15:1	45 days	Serum	15:1	+	
				90 days		15:1	+	
Sea water	*Oreochromis* sp.	Cornmeal + molasses (120 fish/m^3^)	15:1	7 weeks	Plasma	15:1	+	[13]
		Cornmeal + molasses (240 fish/m^3^)				15:1	-	
Triglyceride								
Freshwater	*Cyprinus carpio*	Sugar (6 kg/m^3^)	15:1	49 days	Serum	-	×	[32]
		Sugar (12 kg/m^3^)				15:1	+	
		Rice bran (4.5 kg/m^3^)	20:1	60 days	Serum	-	×	[33]
		Sugarcane molasses (4.5 kg/m^3^)				-	×	
		Rice bran + sugarcane molasses (4.5 kg/m^3^)				-	×	
		Sugar beet molasses	20:1	10 weeks	Serum	-	×	[44]
		Sugar				-	×	
		Corn starch				-	×	
	*Clarias gariepinus*	Sucrose	15:1	6 weeks	Plasma	-	×	[20]
		Glycerol				15:1	+	
		Rice bran				-	×	
Aspartate aminotransferase (AST)								
Freshwater	*Cyprinus carpio*	Sugar (6 kg/m^3^)	15:1	49 days	Serum	15:1	-	[32]
		Sugar (12 kg/m^3^)				15:1	-	
	*Clarias gariepinus*	Sucrose	15:1	6 weeks	Plasma	-	×	[20]
		Glycerol				-	×	
		Rice bran				-	×	
	*Opsariichthys kaopingensis*	Glucose	15:1, 20:1, 25:1	28 days	Serum	15:1, 20:1, 25:1	-	[68]
Sea water	*Paralichthys olivaceus*	Glucose	<10:1	4 months	Plasma	<10:1	-	[28]
Alanine aminotransminase (ALT)								
Freshwater	*Cyprinus carpio*	Sugar (6 kg/m^3^)	15:1	49 days	Serum	15:1	-	[32]
		Sugar (12 kg/m^3^)				15:1	-	
	*Clarias gariepinus*	Sucrose	15:1	6 weeks	Plasma	-	×	[20]
		Glycerol				-	×	
		Rice bran				-	×	
	*Opsariichthys kaopingensis*	Glucose	15:1, 20:1, 25:1	28 days	Serum	20:1	-	[68]
Sea water	*Paralichthys olivaceus*	Glucose	<10:1	4 months	Plasma	<10:1	-	[28]
Alkaline phosphatase (ALP)								
Freshwater	*Oreochromis niloticus*	Sucrose	>10:1	87 days	Serum	-	×	[25]
	*Cyprinus carpio*	Sugar (6 kg/m^3^)	15:1	49 days	Serum	-	×	[32]
		Sugar (12 kg/m^3^)				15:1	+	
Seawater	*Paralichthys olivaceus*	Glucose	<10:1	4 months	Plasma	-	×	[28]
Cortisol								
Freshwater	*Oreochromis niloticus*	Wheat flour (+35% crude protein)	8.4:1	12 weeks	Plasma	-	×	[22]
		Glycerol (140 fish/m^3^)	15:1	98 days	Serum	15:1	-	[31]
		Glycerol (280 fish/m^3^)				15:1	-	
		Molasses (140 fish/m^3^)	15:1	98 days	Serum	15:1	-	
		Molasses (280 fish/m^3^)				15:1	-	
		Starch (140 fish/m^3^)	15:1	98 days	Serum	15:1	-	
		Starch (280 fish/m^3^)				15:1	-	
		Wheat flour (200 fish/m^3^)	15:1	90 days	Serum	15:1	-	[63]
		Wheat flour (250 fish/m^3^)				15:1	-	
		Wheat flour (300 fish/m^3^)				15:1	-	
		Wheat flour (350 fish/m^3^)				15:1	-	
	*Labeo rohita*	Tapioca	15:1	20 days	Serum	15:1	-	[45]
				40 days		-	×	
				60 days		15:1	-	
		Wheat	15:1	20 days	Serum	15:1	-	
				40 days		15:1	-	
				60 days		15:1	-	
		Corn	15:1	20 days	Serum	15:1	-	
				40 days		15:1	-	
				60 days		15:1	+	
		Sugar bagasse	15:1	20 days	Serum	15:1	-	
				40 days		-	×	
				60 days		15:1	-	
	*Cyprinus carpio*	Sugar (6 kg/m^3^)	15:1	49 days	Serum	-	×	[32]
		Sugar (12 kg/m^3^)				15:1	-	
		Corn starch	15:1	60 days	Plasma	15:1	-	[34]
	*Pangasianodon hypophthalmus*	Tapioca	15:1	90 days	Serum	15:1	-	[55]
		Sorghum				15:1	-	
		Pearl millet				15:1	-	
		Finger millet				15:1	-	
Brackish water	*Mugil cephalus*	Sucrose	15:1	60 days	Serum	-	×	[21]
	*Chanos chanos*	Sorghum	15:1	45 days	Serum	15:1	-	[58]
				90 days		15:1	-	
		Potato	15:1	45 days	Serum	15:1	-	
				90 days		15:1	-	
		Yam	15:1	45 days	Serum	15:1	-	
				90 days		15:1	-	
		Glucose	15:1	45 days	Serum	15:1	-	
				90 days		15:1	-	

+, increased, - decreased, and × no change in survival rate.

## 4. Antioxidant Responses

Antioxidant responses in fish, including both enzymatic and non-enzymatic, are closely connected to fish health status, and several types of antioxidant responses are required to control the fish’s complex immune system [75]. Bacterial and viral infections, as well as physical and chemical environmental stress, are the main generators of reactive oxygen species (ROS) in fish, and excessive ROS alters the structural and functional molecules of fish cells, leading to tissue and organ dysfunction by lipid peroxidation [76]. This also induces apoptosis, DNA hydroxylation, protein denaturation, and cell injury [77]. Antioxidant reactions effectively remove the excessively generated ROS as a protection mechanism for the fish, and the excessive ROS exceeding the antioxidant capacity causes oxidative stress [78].

The antioxidant responses of fish species raised in biofloc are shown in Table 4. Total antioxidant capacity (TAC) is an index that measures the antioxidant capacity of all fish, indicating free-radical scavenging ability [68]. Bakhshi et al. [35] reported a significant increase in serum TAC of *C. carpio* cultured with biofloc made from various carbon sources such as sugar beet molasses, sugar, and corn starch. Yu et al. [68,79] reported a significant increase in TAC in various tissues such as the gills, kidney, brain, liver, gut, and serum of *O. kaopingensis* and *C. auratus* cultured with biofloc, and they suggest that antioxidant responses can be increased by bioactive substances such as chlorophyll, polyphenols, carotene, taurine, polysaccharides, phytosterol, and vitamins contained in biofloc, thereby increasing the resistance of fish to environmental stress by lowering the level of lipid peroxidation and inducing a stronger ability in fish to resist free radicals. Yu et al. [80] reported a significant increase in the liver and intestine TAC of Northern snakehead, *Channa argus*, cultured with biofloc, showing that biofloc strengthened antioxidant enzyme activity and relieved oxidative stress.

Superoxide dismutase (SOD) is a major antioxidant enzyme in fish that converts superoxide anion (O_2_^•−^) into hydrogen peroxide to protect fish from damage from reactive oxygen species and maintain the metabolic balance of ROS as a first defense mechanism against oxidative stress [81]. Catalase (CAT) is an enzyme derived from peroxisomes and mitochondria. It establishes a primary antioxidant defense mechanism by converting hydrogen peroxide into water and oxygen with SOD. Mansour and Esteban [28] reported a significant increase in the plasma SOD and CAT activities of *O. niloticus* cultured with biofloc, and they suggest that these results reflect increased fish well-being and reduced oxidative stress. Shourbela et al. [47] observed an increase in serum SOD and CAT activities in *O. niloticus* cultured with biofloc, and within this, the biofloc group with low stocking density showed a significant increase in activities compared to biofloc with high stocking density, mirroring the results of [41]. Menaga et al. [73] suggest that biofloc could cause an increase in SOD and CAT activities in fish and that low levels of SOD and CAT activities meant that high levels of free radicals could accumulate in cells, leading to cell damage. Ebrahimi et al. [33] also reported a significant increase in the serum SOD and CAT activities of *C. carpio* cultured with biofloc, and they suggest that an increase in the antioxidant enzyme that prevents lipid peroxidation improves the antioxidant capacity. Yu et al. [68,79] reported an increase in the SOD and CAT activities of *O. kaopingensis* and *C. auratus* cultured with biofloc, which led to lower levels of lipid peroxidation and a stronger ability to resist free radicals in fish. Nageswari et al. [55] reported a significant increase in the SOD and CAT activities of *P. hypophthalmus* cultured with biofloc, implying that the biofloc environment acts as an effective antioxidant for fish by conferring high resistance to oxidative stress. On the other hand, Haridas et al. [30] reported a decrease in the liver tissue SOD and CAT activities of *O. niloticus* cultured with biofloc, and they suggest that the presence of bioactive compounds may have reduced the production of SOD and CAT. Sontakke et al. [58] reported that the liver SOD and CAT activities of *C. chanos* cultured with biofloc were significantly reduced, suggesting that the antioxidant enzymes were less stimulated due to the lower amount of oxidative stress.

Glutathione peroxide (GPx) plays a critical role in converting hydrogen peroxide into water and oxygen along with CAT, detoxifying its active metabolites and maintaining the intracellular redox balance, thereby protecting fish from cell membrane damage [80]. GPx is a family of enzymes displaying peroxidase activity and a broad substrate spectrum. The enzyme uses glutathione (GSH) as an essential cofactor to catalyze hydrogen peroxide, organic hydrogen peroxide, and lipid hydrogen peroxide with water or alcohol to protect fish from oxidative stress, primarily as an intracellular antioxidant enzyme [78]. Long et al. [29] reported a significant increase in serum GPx of *O. niloticus* cultured with biofloc. Many authors similarly reported a significant increase in GPx in *C. carpio* cultured with biofloc, and they suggest that this increase in the GPx reflects improved fish health and reduced oxidative stress [33,35,63]. Yu et al. [68,79] reported a significant increase in GPx of *O. kaopingensis*, *C. auratus,* and *C. argus* cultured with biofloc, and they suggest that this increase in the GPx induces a stronger ability to resist free radicals. Glutathione reductase (GR) is a component that plays an important role in non-enzymatic antioxidant defense and is a key component of the apoptosis system, converting GSH to glutathione disulfide (GSSG) and back into GSH to stimulate GPx [82]. Shourbela et al. [47] reported a significant increase in the GR of *O. niloticus* cultured with biofloc, and they suggest that the increase in the GR was induced by antioxidant response stimulation. GSH is the first and most important non-enzymatic antioxidant defense mechanism against ROS, including singlet oxygen, superoxide, and hydroxyl radicals in addition to playing a critical role in cell protection, protein synthesis, and cell differentiation and death, which is abundant in the cytoplasm and mitochondria of cells [83]. GSH in fish is essential for assessing redox homeostasis and detoxification conditions in cells with respect to their protective role against oxidative and free-radical-mediated cellular damage [84]. Liu et al. [41,42] reported that GSH of *O. niloticus* cultured with biofloc was significantly increased, which showed that biofloc had an anti-stress effect.

Malondialdehyde (MDA), an important indicator for judging oxidative stress, is the final product of lipid peroxidation produced by the reaction of free radicals with polyunsaturated fatty acids [85]. Liu et al. [41,42] reported a significant decrease in the MDA of *O. niloticus* cultured with biofloc, indicating that fish raised in biofloc had adequate defense against lipid peroxidation and had improved antioxidant capacity and regulatory mechanisms. In addition, it was found that the MDA of *C. carpio* cultured with biofloc was significantly increased due to improved fish health and a reduction in oxidative stress [33]. These reductions in MDA were reported in a variety of biofloc-cultured fish species such as *O. kaopingensis*, *C. auratus,* and *C. argus*, and the results suggest that the bioactive substances, such as chlorophyll, carotene, polysaccharides, polyphenols, phytosterol, taurine, and vitamins in biofloc act as antioxidants [68,79,80]. In fish, the generation of ROS due to various stresses stimulates primary antioxidant enzymes such as SOD and CAT, and GSH is converted to GSSG to activate GPx and then back to GSH with the GR enzyme (Figure 1). However, ROS that are not properly removed may produce lipid peroxide in the lipid membrane, and MDA may be a final product. Therefore, the major antioxidant responses in fish indicate health and the improvement of antioxidant ability by biofloc.

**Table 4 antioxidants-12-00398-t004:** Antioxidant responses of fish species raised in the biofloc in fish aquaculture.

Species		Carbon Source	C:N Ratio	Period	Target Organs	Response C:N Ratio	Response	Reference
Total antioxidant capacity (TAC)								
Freshwater	*Cyprinus carpio*	Sugar beet molasses	20:1	10 weeks	Serum	20:1	+	[44]
		Sugar				20:1	+	
		Corn starch				20:1	+	
	*Opsariichthys kaopingensis*	Glucose	15:1, 20:1, 25:1	28 days	Gills	20:1, 25:1	+	[68]
					Kidney	15:1, 20:1 25:1	+	
					Brain	20:1	+	
					Liver	20:1	+	
					Gut	15:1, 20:1, 25:1	+	
					Serum	20:1, 25:1	+	
	*Carassius auratus*	Anhydrous glucose	10:1, 15:1, 20:1, 25:1	8 weeks	Gut	15:1, 20:1, 25:1	+	[79]
					Kidney	10:1, 15:1, 20:1, 25:1	+	
					Liver	15:1, 20:1, 25:1	+	
	*Channa argus*	Glucose	10:1, 15:1, 20:1	8 weeks	Liver	15:1	+	[80]
					Intestine	15:1	+	
Supero×ide dismutase (SOD)								
Freshwater	*Oreochromis niloticus*	Wheat-milling by-product	15:1	10 weeks	Plasma	15:1	+	[28]
		Rice bran				15:1	+	
		Glycerol (140 fish/m^3^)	15:1	98 days	Serum	15:1	+	[47]
		Glycerol (280 fish/m^3^)				15:1	+	
		Molasses (140 fish/m^3^)	15:1	98 days	Serum	-	×	
		Molasses (280 fish/m^3^)				15:1	+	
		Starch (140 fish/m^3^)	15:1	98 days	Serum	-	×	
		Starch (280 fish/m^3^)				-	×	
		Glucose (166 organisms/m^3^)	15:1	120 days	Liver	15:1	+	[41]
		Glucose (333 organisms/m^3^)				15:1	+	
		Glucose (600 organisms/m^3^)				-	×	
		Wheat flour (200 fish/m^3^)	15:1	90 days	Liver	15:1	-	[30]
		Wheat flour (250 fish/m^3^)				15:1	-	
		Wheat flour (300 fish/m^3^)				15:1	-	
		Wheat flour (350 fish/m^3^)				15:1	-	
		Glucose	10:1, 15:1, 20:1	120 days	Liver	10:1, 15:1	+	[42]
						20:1	-	
		Spentwash	10:1	180 days	Serum	10:1	+	[73]
		Molasses	14:1, 17:1, 20:1	62 days	Liver	-	×	[86]
	*Cyprinus carpio*	Sugar beet molasses	20:1	10 weeks	Serum	-	×	[44]
		Sugar				-	×	
		Corn starch				-	×	
		Sugar (6 kg/m^3^)	15:1	49 days	Serum	-	×	[32]
		Sugar (12 kg/m^3^)				-	×	
		Rice bran	20:1	60 days	Serum	-	×	[33]
		Sugarcane molasses				20:1	+	
		Rice bran + sugarcane molasses				20:1	+	
	*Opsariichthys kaopingensis*	Glucose	15:1, 20:1, 25:1	28 days	Gills	20:1	+	[68]
					Kidney	15:1, 20:1 25:1	+	
					Brain	15:1, 20:1	+	
					Liver	20:1	+	
					Liver	15:1, 25:1	-	
					Gut	15:1, 20:1, 25:1	+	
					Serum	15:1, 20:1, 25:1	+	
	*Carassius auratus*	Anhydrous glucose	10:1, 15:1, 20:1, 25:1	8 weeks	Gut	10:1, 15:1, 20:1, 25:1	+	[79]
					Kidney	15:1, 20:1, 25:1	+	
					Liver	15:1, 20:1, 25:1	+	
	*Pangasianodon hypophthalmus*	Tapioca	15:1	90 days	Serum	15:1	+	[55]
		Sorghum				15:1	+	
		Pearl millet				15:1	+	
		Finger millet				15:1	+	
	*Channa argus*	Glucose	10:1, 15:1, 20:1	8 weeks	Liver	15:1, 20:1	+	[80]
					Intestine	10:1, 15:1, 20:1	+	
Brackish water	*Chanos chanos*	Sorghum	15:1	45 days	Liver	15:1	-	[58]
				90 days		-	×	
		Potato	15:1	45 days	Liver	-	×	
				90 days		-	×	
		Yam	15:1	45 days	Liver	15:1	-	
				90 days		15:1	-	
		Glucose	15:1	45 days	Liver	15:1	-	
				90 days		-	×	
Seawater	*Oreochromis niloticus sp.*	Cornmeal + molasses (120 fish/m^3^)	15:1	7 weeks	Liver	15:1	+	[13]
		Cornmeal + molasses (240 fish/m^3^)				-	×	
Catalase (CAT)								
Freshwater	*Oreochromis niloticus*	Wheat-milling by-product	15:1	10 weeks	Plasma	15:1	+	[28]
		Rice bran				15:1	+	
		Glycerol (140 fish/m^3^)	15:1	98 days	Serum	15:1	+	[47]
		Glycerol (280 fish/m^3^)				15:1	+	
		Molasses (140 fish/m^3^)	15:1	98 days	Serum	-	×	
		Molasses (280 fish/m^3^)				15:1	+	
		Starch (140 fish/m^3^)	15:1	98 days	Serum	-	×	
		Starch (280 fish/m^3^)				15:1	+	
		Wheat flour (200 fish/m^3^)	15:1	90 days	Liver	15:1	-	[63]
		Wheat flour (250 fish/m^3^)				15:1	-	
		Wheat flour (300 fish/m^3^)				15:1	-	
		Wheat flour (350 fish/m^3^)				15:1	-	
		Spentwash	10:1	180 days	Serum	10:1	+	[73]
		Molasses	14:1, 17:1, 20:1	62 days	Liver	-	×	[86]
		Molasses (15% food reduction) (500 fish/m^3^)	15:1	53 days	Skin mucus	-	×	[87]
		Molasses (30% food reduction) (500 fish/m^3^)				15:1	-	
		Molasses (45% food reduction) (500 fish/m^3^)				15:1	-	
		Molasses (100% food reduction) (500 fish/m^3^)				15:1	-	
		Molasses (15% food reduction) (1000 fish/m^3^)	15:1	53 days	Skin mucus	-	×	
		Molasses (30% food reduction) (1000 fish/m^3^)				15:1	-	
		Molasses (45% food reduction) (1000 fish/m^3^)				15:1	-	
		Molasses (100% food reduction) (1000 fish/m^3^)				15:1	-	
	*Cyprinus carpio*	Sugar (6 kg/m^3^)	15:1	49 days	Serum	15:1	+	[32]
		Sugar (12 kg/m^3^)				15:1	-	
		Rice bran	20:1	60 days	Serum	20:1	+	[33]
		Sugarcane molasses				20:1	+	
		Rice bran + sugarcane molasses				20:1	+	
	*Opsariichthys kaopingensis*	Glucose	15:1, 20:1, 25:1	28 days	Gills	20:1	+	[68]
					Kidney	20:1, 25:1	+	
					Brain	15:1, 20:1, 25:1	+	
					Liver	20:1, 25:1	+	
					Gut	15:1, 20:1, 25:1	+	
					Serum	20:1, 25:1	+	
	*Carassius auratus*	Anhydrous glucose	10:1, 15:1, 20:1, 25:1	8 weeks	Gut	10:1, 15:1, 20:1, 25:1	+	[79]
					Kidney	15:1, 20:1, 25:1	+	
					Liver	10:1, 15:1, 20:1, 25:1	+	
	*Pangasianodon hypophthalmus*	Tapioca	15:1	90 days	Serum	15:1	+	[55]
		Sorghum				15:1	+	
		Pearl millet				15:1	+	
		Finger millet				15:1	+	
	*Channa argus*	Glucose	10:1, 15:1, 20:1	8 weeks	Liver	10:1, 15:1, 20:1	+	[80]
					Intestine	15:1, 20:1	+	
Brackish water	*Chanos chanos*	Sorghum	15:1	45 days	Liver	15:1	-	[58]
				90 days		-	×	
		Potato	15:1	45 days	Liver	15:1	-	
				90 days		-	×	
		Yam	15:1	45 days	Liver	15:1	-	
				90 days		15:1	-	
		Glucose	15:1	45 days	Liver	15:1	-	
				90 days		15:1	-	
Seawater	*Oreochromis niloticus* sp.	Cornmeal + molasses (120 fish/m^3^)	15:1	7 weeks	Liver	-	×	[13]
		Cornmeal + molasses (240 fish/m^3^)				-	×	
Glutathione peroxidase (GPx)								
Freshwater	*Oreochromis niloticus*	Glucose	15:1	8 weeks	Serum	15:1	+	[29]
	*Cyprinus carpio*	Sugar beet molasses	20:1	10 weeks	Serum	20:1	+	[44]
		Sugar				20:1	-	
		Corn starch				20:1	-	
		Sugar (6 kg/m^3^)	15:1	49 days	Serum	15:1	+	[32]
		Sugar (12 kg/m^3^)				15:1	-	
		Rice bran	20:1	60 days	Serum	-	×	[33]
		Sugarcane molasses				-	×	
		Rice bran + sugarcane molasses				20:1	+	
	*Opsariichthys kaopingensis*	Glucose	15:1, 20:1, 25:1	28 days	Gills	20:1, 25:1	+	[68]
					Kidney	20:1, 25:1	+	
					Brain	15:1, 20:1, 25:1	+	
					Liver	20:1	+	
					Gut	20:1	+	
					Serum	20:1, 25:1	+	
	*Carassius auratus*	Anhydrous glucose	10:1, 15:1, 20:1, 25:1	8 weeks	Gut	10:1, 15:1, 20:1, 25:1	+	[79]
					Kidney	10:1, 15:1, 20:1, 25:1	+	
					Liver	15:1, 20:1, 25:1	+	
	*Channa argus*	Glucose	10:1, 15:1, 20:1	8 weeks	Liver	15:1,	+	[80]
					Intestine	10:1, 15:1, 20:1	+	
Seawater	*Oreochromis niloticus* sp.	Cornmeal + molasses (120 fish/m^3^)	15:1	7 weeks	Liver	-	×	[13]
		Cornmeal + molasses (240 fish/m^3^)				-	×	
Glutathione reductase (GR)								
Freshwater	*Oreochromis niloticus*	Glycerol (140 fish/m^3^)	15:1	98 days	Serum	15:1	+	[47]
		Glycerol (280 fish/m^3^)				15:1	+	
		Molasses (140 fish/m^3^)	15:1	98 days	Serum	-	×	
		Molasses (280 fish/m^3^)				15:1	+	
		Starch (140 fish/m^3^)	15:1	98 days	Serum	-	×	
		Starch (280 fish/m^3^)				-	×	
Seawater	*Oreochromis niloticus* sp.	Cornmeal + molasses (120 fish/m^3^)	15:1	7 weeks	Liver	-	×	[13]
		Cornmeal + molasses (240 fish/m^3^)				-	×	
Reduced glutathione (GSH)								
Freshwater	*Oreochromis niloticus*	Glucose (166 organisms/m^3^)	15:1	120 days	Liver	15:1	+	[41]
		Glucose (333 organisms/m^3^)				15:1	+	
		Glucose (600 organisms/m^3^)				-	×	
		Glucose	10:1, 15:1, 20:1	120 days	Liver	10:1, 15:1	+	[42]
Malondialdehyde (MDA)								
Freshwater	*Oreochromis niloticus*	Glucose (166 organisms/m^3^)	15:1	120 days	Liver	15:1	+	[41]
		Glucose (333 organisms/m^3^)				15:1	+	
		Glucose (600 organisms/m^3^)				-	×	
		Glucose	10:1, 15:1, 20:1	120 days	Liver	10:1, 15:1	+	[42]
	*Cyprinus carpio*	Sugar (6 kg/m^3^)	15:1	49 days	Serum	15:1	+	[72]
		Sugar (12 kg/m^3^)				15:1	-	
		Rice bran	20:1	60 days	Serum	-	×	[33]
		Sugarcane molasses				20:1	-	
		Rice bran + sugarcane molasses				-	×	
	*Clarias gariepinus*	Sucrose	15:1	6 weeks	Muscle	-	×	[20]
		Glycerol				-	×	
		Rice bran				-	×	
	*Opsariichthys kaopingensis*	Glucose	15:1, 20:1, 25:1	28 days	Gills	-	×	[68]
					Kidney	20:1	-	
					Brain	-	×	
					Liver	20:1, 25:1	-	
					Gut	20:1	-	
					Serum	15:1, 20:1, 25:1	-	
	*Carassius auratus*	Anhydrous glucose	10:1, 15:1, 20:1, 25:1	8 weeks	Gut	20:1, 25:1	-	[79]
					Kidney	20:1, 25:1	-	
					Liver	15:1, 20:1, 25:1	-	
	*Channa argus*	Glucose	10:1, 15:1, 20:1	8 weeks	Liver	15:1	-	[80]
					Intestine	10:1, 15:1, 20:1	-	
Seawater	*Oreochromis niloticus* sp.	Cornmeal + molasses (120 fish/m^3^)	15:1	7 weeks	Liver	-	×	[13]
		Cornmeal + molasses (240 fish/m^3^)				-	×	

+, increased, - decreased, and × no change in survival rate.

## 5. Immune Responses

The effective microorganisms in biofloc act as immune- and growth-stimulating factors, and the improvement in the immunity of the fish due to biofloc may have a mitigating effect against fatal pathogenic infections [88]. Biofloc carotenoids are known to carry out various bioactive physiological functions including stimulating the fish immune system and providing essential nutrition, and the cell wall components (β-1,3-glucans, LPS, and peptidoglycan) of microorganisms present in biofloc improve the innate immune response of fish [89,90]. Ekasari et al. [10] also found that the consumption of biofloc can induce non-specific immune system stimulation through continuous exposure to microbe-associated molecular patterns (MAMPs) such as β-1,3,-glucans, lipopolysaccharides, and peptidoglycan.

The immune responses of fish raised in the biofloc are demonstrated in Table 5. Macrophages are one of the most primitive phagocytic cells in the non-specific immune system of fish, and macrophage phagocytosis, the process by which a cell engulfs foreign substances (>0.5 μm) into endocytic vesicles named phagosomes, is a major indicator in assessing immune function under various biotic and abiotic factors (including contaminants, pathogens, and genetic variation) [91]. Phagocytic processes include foreign substance detection and recognition, foreign body attachment to phagocytes, foreign particle engulfment or internalization into phagosomes, the fusion of phagosomes with a lysosome, and formation of phagolysosomes (through degranulation of the phagocyte and maturation of compartments through endosomal fusion). Intracellular death and digestion of foreign particles and some phagocyte (dendritic cells and macrophages) uptake and antigen presentation [92] can also occur. Mansour and Esteban [28] reported a significant increase in macrophage phagocytosis of *O. niloticus*, and this increase indicates an improvement in the innate immune status of fish in biofloc. Fauji et al. [19] also reported a significant increase in phagocytosis in *C. gariepinus*, and they also suggest that it indicates higher immunity and better health status of fish cultured with biofloc.

Immunoglobulins are major immune molecules that recognize antigens, and they have critical roles in the host destruction of antigens and adaptive immune response responsible for fish immune memory [93]. In addition, immunoglobulins are the main humoral components of specific immunity, and total immunoglobulin concentrations are closely related to fish physiology and pathology [94,95]. Six immunoglobulin isotypes (Ig M, Ig G, Ig A, Ig E, Ig D, and Ig O) have been identified in mammals, but three isotypes (Ig M, Ig D, and Ig T/Z) have been reported so far in teleost [96]. Mansour and Esteban [28] reported a significant increase in total immunoglobulin of *O. niloticus* cultured with biofloc, and they suggest that this increase was due to the improved immunity provided by the bioactive substances. Verma et al. [45] reported a significant increase in total immunoglobulin in *L. rohita*, and Bakhshi et al. [35] also reported a significant increase in total immunoglobulin of *C. carpio*, which was argued to be a result of the components of biofloc having a positive effect on immune parameters by altering the intestinal microbiome, thereby increasing the concentration of immunoglobulins in the serum. Immunoglobulin M (Ig M) is the only component in certain humoral defense systems and plays a critical role in determining and neutralizing foreign antigens such as pathogenic bacteria and viruses [97]. Teleost Ig M is similar to mammalian Ig M in physiological properties, structure, soluble form, and membrane-bound form, and it plays an important role as an immune effector molecule in fish blood as well as fish skin [98]. Kishawy et al. [24] reported a significant increase in Ig M in *O. niloticus* cultured with biofloc, and the increase was associated with the immunostimulating effect of β-1,3,-glucans, LPS, and peptidoglycans in the cell wall of probiotic bacteria present in biofloc. Ebrahimi et al. [33] reported a significant increase in Ig M in *C. carpio* cultured with biofloc, which means that the carbon source of biofloc and the energy provided by microbial flocs are effective not only for tissue growth and body maintenance but also for stimulating fish immunity. Yu et al. [80] reported a significant increase in Ig M in *C. argus*, and Kim et al. [27] also reported a similar significant increase in *P. olivaceus*, which together indicate that biofloc can act as an immunostimulant.

Lysozyme is an enzyme produced by leukocytes in the blood that lyses bacterial cell walls and stimulates phagocytosis, therefore fighting pathogenic infection and disease [58]. Lysozyme is the first line of defense of fish immunity against various pathogens including bacteria, viruses, and parasites, and its activity in fish can be altered by health, sex, environmental stress, and toxic substances [43]. Mansour and Esteban [28] reported a significant increase in lysozyme in *O. niloticus* cultured with biofloc, suggesting an improvement in immunity. Many authors have reported this increase, and it has been argued that bioactive compounds that are present in biofloc, such as natural microorganisms, carotenoids, and fat-soluble vitamins, stimulate the fish’s immune response [24,29,30,41,43]. Verma et al. [45] reported a significant increase in lysozyme activity in *L. rohita* cultured with biofloc, suggesting that breeding fish in a biofloc-based system can induce the enhancement of non-specific immunity. Yu et al. [68,79,80] reported an increase in the activity of lysozyme in many fish species such as *O. kaopingensis*, *C. auratus,* and *C. argus* cultured with biofloc, which means that microbial flocs could be a protein source for the fish immune response.

Myeloperoxidase (MPO), a ferrous lysosomal protein in the myeloid cells of monocytes and neutrophils, is the most abundantly expressed peroxidase in neutrophil granulocytes and liberates hypochlorous acid (HOCl) from hydrogen peroxide (H_2_O_2_) and chloride anions (Cl^−^) during the respiratory burst of neutrophils [58]. MPO is widely distributed in the liver, head-kidney, heart, muscle, and intestinal tissues of teleost due to the spread of neutrophil granulocytes, which are highly expressed in head-kidney and spleen tissues [99]. MPO is most abundant in neutrophil primary granules and fuses with phagocyte vesicles to accelerate pathogen destruction, thereby delivering these compounds to invading pathogens. It produces high levels of HOCl to deliver a strong antibacterial response [100]. Many authors reported a significant increase in the MPO of *O. niloticus* cultured with biofloc, and they suggest that this increase was due to the stimulation or activation of the fish immune system [28,30,73]. Verma et al. [45] found there was an increase in MPO of *L. rohita* cultured with biofloc, but the degree of stimulation was different depending on the carbon source that produced the biofloc. Although there was a significant decrease in MPO in biofloc made with some carbon sources, other carbon sources showed a significant increase, implying that they have high antibacterial activity through hypochlorous acid production during respiratory bursts. Respiratory burst, a measure of oxygen-dependent defense mechanisms in fish phagocytic cells, is a reaction that occurs in phagocytes to degrade pathogenic bacteria and internalized particles as part of a vital immune response, and the increase in respiratory burst may correlate with increased apoptotic activity against pathogenic bacteria and internalized particles, as the ability of macrophages to kill pathogenic microbes is an important mechanism to protect them against disease in fish [58,101]. Leukocyte respiratory burst activity is measured because phagocytosis was associated with increased oxygen consumption, with the activity also being closely related to inflammatory responses and cytokine release in fish [102]. Mansour and Esteban [28] reported that the respiratory burst activity of *O. niloticus* cultured with biofloc was significantly increased, which is considered to be due to the stimulation of biofloc-induced fish cell defenses. Haridas et al. [30] reported the same respiratory burst activity and considered it to be a phenomenon of the natural probiotic effect in biofloc. Verma et al. [45] reported an increase in respiratory burst activity of *L. rohita* cultured with biofloc, and the increase differed depending on the biofloc environment and the carbon source, with the highest respiratory rupture activity being observed in the biofloc made from tapioca. Sontakke et al. [58] reported an increase in respiratory burst activity of *C. chanos* cultured with biofloc; a result of it acting as an immunostimulant.

This effect is achieved via three pathways, including alternative (independent of the antibody and stimulated directly by bacteria, fungi, viruses, or tumor cells), lectin (stimulated through the binding of ficolins and collectins to carbohydrates present on the surfaces of pathogens) and classical (stimulated when the pattern recognition molecule C1q binds to the CH_3_ domain(IgM) or CH2 domain(IgG) complexed to antigens). The ACH_50_ assay (Alternative Complement pathway Hemolytic activity) index is generally used for the analysis of alternative complement activity, and it calculates the volume of serum in sheep or rabbits that is 50% lysed [103,104]. ACH_50_ is an innate immune system reaction, associated with opsonization, inflammation, and cell membrane attack [105]. Many authors have reported a significant increase in serum ACH_50_ of *O. niloticus* and *C. carpio* cultured with biofloc, meaning that bioactive compounds such as chlorophylls, short-chain fatty acids, amino sugars, phytosterols, bromophenols, carotenoids, and anti-bacterial compounds like poly-β-hydroxybutyrate present in biofloc induce stimulation in fish immunity [28,32,33]. Complement 3 (C3), a critical humoral component in the innate immune response, has a critical function in alerting the host immune system to the presence and clearance of potential pathogens. It is connected to all three pathways that directly lyse pathogenic cells by merging and proceeding through terminal pathways leading to the formation of membrane attack complexes (MAC) [81]. C4 is also a critical humoral component in the innate immune response, and it plays an activating role in the process of MAC formation in the classical and lectin pathway. The covalent tagging of the foreign molecules by C3 or C4 is an important process in complement stimulation, which results in phagocytosis via binding to the complement receptors on phagocytes, or cytolysis via the stimulation of late components [106] Many authors reported a significant increase in C3 in *O. niloticus* and *C. argus* cultured with biofloc, and they suggest that this increase indicates an improved immune status in fish [41,42,80].

**Table 5 antioxidants-12-00398-t005:** Immune responses of fish species raised in the biofloc in fish aquaculture.

Species		Carbon Source	C:N Ratio	Period	Target Organs	Response C:N Ratio	Response	Reference
Phagocytosis								
Freshwater	*Oreochromis niloticus*	Wheat-milling by-product	15:1	10 weeks	Macrophages	15:1	+	[28]
		Rice bran				15:1	+	
	*Clarias gariepinus*	Tapioca flour (4 fish/L)	10:1	20 days	Blood	-	×	[19]
		Tapioca flour (6 fish/L)				-	×	
		Tapioca flour (8 fish/L)				10:1	+	
Total immunoglobulin								
Freshwater	*Oreochromis niloticus*	Wheat-milling by-product	15:1	10 weeks	Plasma	15:1	+	[28]
		Rice bran				15:1	+	
		Molasses (15% food reduction) (500 fish/m^3^)	15:1	53 days	Skin mucus	-	×	[87]
		Molasses (30% food reduction) (500 fish/m^3^)				-	×	
		Molasses (45% food reduction) (500 fish/m^3^)				-	×	
		Molasses (100% food reduction) (500 fish/m^3^)				15:1	-	
		Molasses (15% food reduction) (1000 fish/m^3^)	15:1	53 days	Skin mucus	-	×	
		Molasses (30% food reduction) (1000 fish/m^3^)				-	×	
		Molasses (45% food reduction) (1000 fish/m^3^)				-	×	
		Molasses (100% food reduction) (1000 fish/m^3^)				-	×	
	*Labeo rohita*	Tapioca	15:1	20 days	Plasma	-	×	[45]
				40 days		15:1	+	
				60 days		15:1	+	
		Wheat	15:1	20 days	Plasma	-	×	
				40 days		15:1	+	
				60 days		15:1	+	
		Corn	15:1	20 days	Plasma	-	×	
				40 days		15:1	+	
				60 days		-	×	
		Sugar bagasse	15:1	20 days	Plasma	15:1	-	
				40 days		15:1	+	
				60 days		-	×	
	*Cyprinus carpio*	Sugar beet molasses	20:1	10 weeks	Serum	20:1	+	[44]
		Sugar				20:1	+	
		Corn starch				20:1	+	
Immunoglobulin M (IgM)								
Freshwater	*Oreochromis niloticus*	Glycerol	15:1	12 weeks	Serum	15:1	+	[24]
		Mannan oligosaccharides				15:1	+	
	*Cyprinus carpio*	Rice bran	20:1	60 days	Serum	20:1	+	[33]
		Sugarcane molasses				20:1	+	
		Rice bran + sugarcane molasses				20:1	+	
	*Channa argus*	Glucose	10:1, 15:1, 20:1	8 weeks	Serum	-	×	[80]
					Kidney	15:1, 20:1	+	
Seawater	*Paralichthys olivaceus*	Glucose	10:1	4 months	Plasma	10:1	+	[28]
Lysozyme activity								
Freshwater	*Oreochromis niloticus*	Wheat-milling by-product	15:1	10 weeks	Plasma	15:1	+	[28]
		Rice bran				15:1	+	
		Sucrose	>10:1	87 days	Hepatopancreas	-	×	[25]
					Head kidney	-	×	
					Serum	-	×	
		Glucose	15:1	8 weeks	Serum	15:1	+	[29]
		Glucose (166 organisms/m^3^)	15:1	120 days	Liver	15:1	+	[41]
		Glucose (333 organisms/m^3^)				15:1	+	
		Glucose (600 organisms/m^3^)				15:1	-	
		Wheat flour (200 fish/m^3^)	15:1	90 days	Serum	15:1	+	[30]
		Wheat flour (250 fish/m^3^)				15:1	+	
		Wheat flour (300 fish/m^3^)				15:1	+	
		Wheat flour (350 fish/m^3^)				15:1	+	
		Glycerol	15:1	12 weeks	Serum	15:1	+	[24]
		Mannan oligosaccharides				15:1	+	
		Glucose	10:1, 15:1, 20:1	120 days	Liver	10:1, 15:1, 20:1	+	[42]
		Molasses (15% food reduction) (500 fish/m^3^)	15:1	53 days	Skin mucus	-	×	[87]
		Molasses (30% food reduction) (500 fish/m^3^)				15:1	-	
		Molasses (45% food reduction) (500 fish/m^3^)				15:1	-	
		Molasses (100% food reduction) (500 fish/m^3^)				15:1	-	
		Molasses (15% food reduction) (1000 fish/m^3^)	15:1	53 days	Skin mucus	-	×	
		Molasses (30% food reduction) (1000 fish/m^3^)				-	×	
		Molasses (45% food reduction) (1000 fish/m^3^)				15:1	-	
		Molasses (100% food reduction) (1000 fish/m^3^)				15:1	-	
	*Labeo rohita*	Tapioca	15:1	20 days	Serum	-	×	[45]
				40 days		15:1	+	
				60 days		15:1	+	
		Wheat	15:1	20 days	Serum	15:1	+	
				40 days		15:1	+	
				60 days		15:1	+	
		Corn	15:1	20 days	Serum	-	×	
				40 days		-	×	
				60 days		15:1	-	
		Sugar bagasse	15:1	20 days	Serum	-	×	
				40 days		-	×	
				60 days		-	×	
	*Cyprinus carpio*	Sugar beet molasses	20:1	10 weeks	Serum	-	×	[44]
		Sugar				-	×	
		Corn starch				-	×	
		Sugar (6 kg/m^3^)	15:1	49 days	Serum	15:1	×	[32]
		Sugar (12 kg/m^3^)				15:1	×	
		Rice bran	20:1	60 days	Serum	20:1	+	[33]
		Sugarcane molasses				-	×	
		Rice bran + sugarcane molasses				-	×	
	*Opsariichthys kaopingensis*	Glucose	15:1, 20:1, 25:1	28 days	Gills	20:1	+	[68]
					Kidney	15:1, 20:1 25:1	+	
					Brain	20:1	+	
					Liver	20:1, 25:1	+	
					Gut	15:1, 20:1, 25:1	+	
	*Carassius auratus*	Anhydrous glucose	10:1, 15:1, 20:1, 25:1	8 weeks	Gut	15:1, 20:1, 25:1	+	[79]
					Kidney	15:1, 20:1, 25:1	+	
					Liver	20:1, 25:1	+	
	*Channa argus*	Glucose	10:1, 15:1, 20:1	8 weeks	Serum	10:1, 15:1, 20:1	+	[80]
					Kidney	15:1, 20:1	+	
Myeloperoxidase (MPO)								
Freshwater	*Oreochromis niloticus*	Wheat-milling by-product	15:1	10 weeks	Blood	15:1	+	[28]
		Rice bran				15:1	+	
		Wheat flour (200 fish/m^3^)	15:1	90 days	Serum	15:1	+	[30]
		Wheat flour (250 fish/m^3^)				15:1	+	
		Wheat flour (300 fish/m^3^)				15:1	+	
		Wheat flour (350 fish/m^3^)				-	×	
		Spentwash	10:1	180 days	Serum	10:1	+	[73]
	*Labeo rohita*	Tapioca	15:1	20 days	Serum	15:1	-	[45]
				40 days		15:1	+	
				60 days		15:1	+	
		Wheat	15:1	20 days	Serum	15:1	-	
				40 days		-	×	
				60 days		15:1	+	
		Corn	15:1	20 days	Serum	15:1	-	
				40 days		-	×	
				60 days		-	×	
		Sugar bagasse	15:1	20 days	Serum	15:1	-	
				40 days		-	×	
				60 days		-	×	
Respiratory burst activity								
Freshwater	*Oreochromis niloticus*	Wheat-milling by-product	15:1	10 weeks	Leucocytes	15:1	+	[28]
		Rice bran				15:1	+	
		Wheat flour (200 fish/m^3^)	15:1	90 days	Phagocytes	15:1	+	[30]
		Wheat flour (250 fish/m^3^)				15:1	+	
		Wheat flour (300 fish/m^3^)				15:1	+	
		Wheat flour (350 fish/m^3^)				15:1	+	
	*Labeo rohita*	Tapioca	15:1	20 days	Leucocytes	15:1	+	[45]
				40 days		15:1	+	
				60 days		15:1	+	
		Wheat	15:1	20 days	Leucocytes	15:1	+	
				40 days		15:1	+	
				60 days		15:1	+	
		Corn	15:1	20 days	Leucocytes	-	×	
				40 days		15:1	+	
				60 days		15:1	+	
		Sugar bagasse	15:1	20 days	Leucocytes	15:1	×	
				40 days		15:1	+	
				60 days		15:1	+	
Brackish water	*Chanos chanos*	Sorghum	15:1	45 days	Phagocytes	15:1	+	[58]
				90 days		15:1	+	
		Potato	15:1	45 days	Phagocytes	15:1	+	
				90 days		15:1	+	
		Yam	15:1	45 days	Phagocytes	15:1	+	
				90 days		15:1	+	
		Glucose	15:1	45 days	Phagocytes	15:1	+	
				90 days		15:1	+	
Alterative complement activity (ACH_50_)								
Freshwater	*Oreochromis niloticus*	Wheat-milling by-product	15:1	10 weeks	Plasma	15:1	+	[28]
		Rice bran				15:1	+	
	*Cyprinus carpio*	Sugar beet molasses	20:1	10 weeks	Serum	-	×	[44]
		Sugar				-	×	
		Corn starch				-	×	
		Sugar (6 kg/m^3^)	15:1	49 days	Serum	15:1	×	[32]
		Sugar (12 kg/m^3^)				15:1	+	
		Rice bran	20:1	60 days	Serum	20:1	+	[30]
		Sugarcane molasses				20:1	+	
		Rice bran + sugarcane molasses				20:1	+	
Complement 3 (C3)								
Freshwater	*Oreochromis niloticus*	Glucose (166 organisms/m^3^)	15:1	120 days	Liver	15:1	+	[41]
		Glucose (333 organisms/m^3^)				15:1	+	
		Glucose (600 organisms/m^3^)				15:1	-	
		Glucose	10:1, 15:1, 20:1	120 days	Liver	15:1, 20:1	+	[42]
	*Channa argus*	Glucose	10:1, 15:1, 20:1	8 weeks	Serum	10:1, 15:1	+	[80]
					Kidney	15:1	+	
Complement 4 (C4)								
*Freshwater*	*Channa argus*	Glucose	10:1, 15:1, 20:1	8 weeks	Serum	10:1, 15:1, 20:1	+	[80]
					Kidney	10:1, 15:1, 20:1	+	

+, increased, - decreased, and × no change in survival rate.

## 6. Disease Resistance

In a biofloc environment, various bioactive substances including chlorophyll, polyphenols, carotene, taurine, polysaccharides, phytosterol, and vitamins have unique antagonistic applications against pathogens, thereby suppressing disease outbreaks and improving the immunity of cultured fish [5]. *Bacillus*, the dominant bacterium in a biofloc environment, is an important probiotic in fish farming, as it enhances the disease resistance and immunity of fish [107]. As effective microorganisms of biofloc compete with pathogenic bacteria, when pathogenic bacteria enter the breeding water, they have limited proliferation when compared to general farms. This is due to competition for dominance with useful microorganisms, thereby reducing damage in fish aquaculture caused by bacterial diseases [73]. These advantages of biofloc mean that antibiotics that affect aquaculture organisms, farm environments, and food hygiene are not used in fish culture using biofloc, and the occurrence of antibiotic-resistant superbacteria caused by the misuse of antibiotics can also be prevented [108]. According to recent studies, biofloc has been proven to have a protective effect against various diseases such as *Vibrio harveyi*, *Aeromonas hydrophila, Edwardsiella tarda,* and *Streptococcus iniae*, all of which cause great damage in fish farms [5,35].

A pathogen challenge test of fish species raised in biofloc in fish aquaculture is shown in Table 6. Liu et al. [41] reported that the survival rate of *V. harveyi*-infected *O. niloticus* cultured with biofloc was higher than that of the control group, which is considered to be the result of immune system stimulation. Kishawy et al. [24] suggest that the survival rate of *O. niloticus* infected with *A. hydrophila* was higher in the groups cultured with biofloc, and they suggest that using MOS as the carbon source in biofloc improved fish immunity and disease resistance, thereby improving the survival rate. Haridas et al. [30] also reported a high survival rate during aa *A. hydrophila* infection in *O. niloticus* cultured with biofloc, which again indicates positive effects on disease resistance. Fauji et al. [19] reported a significant increase in survival rate following infection with *A. hydrophila* in *C. gariepinus* cultured with biofloc. Verma et al. [45] reported that the survival rate from an *A. hydrophila* infection of *L. rohita* reared with biofloc was higher than that of the control group. Bakhshi et al. [35] similarly reported a high survival rate from an *A. hydrophila* infection of *C. carpio* L. cultured with biofloc. Kim et al. [67] reported that the survival rate of *P. olivaceus* cultured with biofloc with an *E. tarda* infection was significantly increased, and this was due to the improvement of disease resistance induced by the immunostimulation of biofloc.

## 7. Conclusions

Fish raised in biofloc can contribute to productivity improvements in the fish culture industry due to the improved growth and high survival rate in fish raised in biofloc. The biofloc-raised fish species demonstrated better physiological indicators and lower stress compared to the control in hematological parameters such as RBC, WBC, Ht, Hb, glucose, cholesterol, total protein, albumin, globulin, triglyceride, AST, ALT, ALP, and cortisol. In antioxidant responses indicated by, e.g., TAC, SOD, CAT, GPx, GR, GSH, and MDA, fish raised in biofloc showed higher antioxidant stimulation, suggesting that they had a stronger ability to remove ROS caused by environmental stress. Immune responses such as phagocytosis, total immunoglobulin, Ig M, lysozyme activity, MPO, respiratory burst activity, ACH_50_, C3, and C4 were stimulated by a significant amount in various fish species. The fish raised in biofloc had higher disease resistance in challenge tests of major fish diseases such as *V. harveyi*, *A. hydrophila, E. tarda,* and *S. iniae*. In conclusion, various physiological effects and conferred disease resistance of biofloc in fish aquaculture have been confirmed. Our results provide important information in identifying fish health, which is directly related to fish production, and can be used as a standard for applying biofloc to fish aquaculture in the future.

## Figures and Tables

**Figure 1 antioxidants-12-00398-f001:**
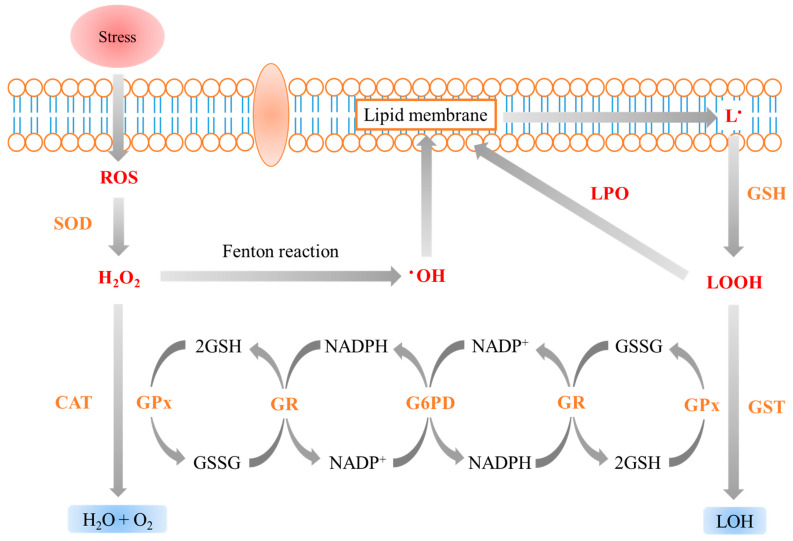
The mechanisms of antioxidant responses such as SOD, CAT, GPx, GR, GSH in fish (^•^OH : hydroxyl radical, L•: lipid radical, LOOH: lipid hydroperoxide, LPO: lipid peroxidation).

**Table 6 antioxidants-12-00398-t006:** Pathogen challenge test of fish species raised in the biofloc in fish aquaculture.

Species		Pathogen (Strain)	CFU	Carbon Source	C/N Ratio	Period	Response C/N Ratio	Response	Reference
Survival rates									
Freshwater	*Oreochromis niloticus*	*Vibrio harveyi*	4 × 10^8^ CFU/mL	Glucose (166 organisms/m^3^)	15:1	14 days	15:1	+	[41]
				Glucose (333 organisms/m^3^)			15:1	+	
				Glucose (600 organisms/m^3^)			-	×	
		*Aeromonas hydrophila* (ATCC 7966)	3 × 10^8^ CFU/mL	Mannan oligosaccharides + plant-protein-based diet	15:1	3 days	15:1	+	[24]
				Mannan oligosaccharides + fish-protein-based diet			15:1	+	
				Glycerol + plant-protein-based diet			15:1	+	
				Glycerol + fish-protein-based diet			15:1	+	
		*Aeromonas hydrophila* (ATCC 7966)	3 × 10^8^ CFU/mL	Mannan oligosaccharides + plant-protein-based diet	15:1	7 days	15:1	+	
				Mannan oligosaccharides + fish-protein-based diet			15:1	+	
				Glycerol + plant-protein-based diet			15:1	+	
				Glycerol + fish-protein-based diet			15:1	+	
		*Aeromonas hydrophila* (ATCC 7966)	10^6^ CFU/mL	Wheat flour (200 fish/m^3^)	15:1	3 days	15:1	+ (83.33%)	[30]
				Wheat flour (250 fish/m^3^)			15:1	+ (83.33%)	
				Wheat flour (300 fish/m^3^)			15:1	+ (75%)	
				Wheat flour (350 fish/m^3^)			15:1	+ (62.5%)	
	*Clarias gariepinus*	*Aeromonas hydrophila*	10^6^ CFU/mL	Tapioca (4 fish/L)	10:1	7 days	10:1	+	[19]
				Tapioca (6 fish/L)			10:1	+	
				Tapioca (8 fish/L)			10:1	+	
	*Labeo rohita*	*Aeromonas hydrophila* (ATCC 7966)	1.8 × 10^7^ CFU/mL	Tapioca	15:1	14 days	15:1	+	[45]
				Wheat			15:1	+	
				Corn			15:1	+	
				Sugar bagasse			15:1	+	
	*Cyprinus carpio*	*Aeromonas hydrophila* (ATCC 897)	1.1 × 10^7^ CFU/mL	Sugar beet molasses	15:1	14 days	15:1	+	[44]
				Sugar			15:1	+ (50%)	
				Corn starch			15:1	+ (50%)	
Seawater	*Paralichthys olivaceus*	*Edwardsiella tarda* (FP 5060)	6.61 × 10^4^ CFU/g fish	-	-	7 days	-	×	[5]
			6.61 × 10^5^ CFU/g fish				-	+ (100%)	
			6.61 × 10^6^ CFU/g fish				-	+ (33%)	
			6.61 × 10^7^ CFU/g fish				-	+ (33%)	
		*Streptococcus iniae* (FP 5228)	3.36 × 10^6^ CFU/g fish	Glucose	10:1	96 h	10:1	+ (100%)	[70]
			3.36 × 10^7^ CFU/g fish				10:1	+ (80%)	
			3.36 × 10^8^ CFU/g fish				10:1	+ (70%)	
			3.36 × 10^9^ CFU/g fish				10:1	× (20%)	

+, increased, - decreased, and × no change in survival rate.

## Data Availability

Not applicable.

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
