# Peer review of "Biofloc Technology in Fish Aquaculture: A Review"

_antioxidants, 2023, doi:10.3390/antiox12020398_

Round 1
Reviewer 1 Report
Dear authors, congratulations for carrying out a topical study in the field of aquaculture, namely about bio-floc technology. The text of the work is transdisciplinary and will be useful both to researchers in the field of aquaculture and to innovative industries in the field. I have only one recommendation, namely to include in the introduction a description/definition of bio-floc technology so that researchers from transdisciplinary fields such as the study of antioxidant compounds can familiarize themselves with the field in this work without looking elsewhere . Aquaculture researchers are familiar with this technology, but those from transdisciplinary fields are not.
Author Response
Reviewers' comments:
Reviewer 1.
Dear authors, congratulations for carrying out a topical study in the field of aquaculture, namely about bio-floc technology. The text of the work is transdisciplinary and will be useful both to researchers in the field of aquaculture and to innovative industries in the field. I have only one recommendation, namely to include in the introduction a description/definition of bio-floc technology so that researchers from transdisciplinary fields such as the study of antioxidant compounds can familiarize themselves with the field in this work without looking elsewhere . Aquaculture researchers are familiar with this technology, but those from transdisciplinary fields are not.
: Thank you for pointing this out. We aggree with the opinions of the reviewer. We revised the sentence and added reference as below.
Original sentence
“BFT, which removes ammonia and nitrite naturally occurring during the aquaculture process due to the fish metabolic activity and uneaten feed waste, is important for a sustainable future in fish aquaculture by using effective microorganisms.”
Revised sentence
“Bio-floc technology (BFT) is an eco-friendly technology that effectively treats ammonia and nitrite caused by feed waste, feces and urine, which are naturally generated in the metabolic process of aquatic products, using beneficial microorganisms [9]. The basic principle of this technology is to perform a self-nitrification process in aquaculture systems without exchange of stock water by stimulating the growth of benefical microorganisms that can be utilized as a feed source for aquaculture species and act as absorbing nitrogenous waste [10].”
English editing is in progress, and it will be reflected in the final submission.
We sincerely thanks for your careful review to improve this article.
Reviewer 2 Report
In the study entitled “Bio-floc technology in fish aquaculture: A review” the Authors aimed to reviewed the hematological parameters, antioxidant and immune responses in fish raised in bio-floc in order to assess the health status and physiological activity of bio-floc on fish. The resistance to diseases of fish raised in bio-floc was evaluated, via a challenge test to major specific diseases of fish. There have been many researches on the productivity increase, physiological activity and disease resistance of bio-floc in fish culture, and this review suggest the standards for applying bio-floc to fish aquaculture. All the aspects discussed in this review are worthy of investigation and, the authors well and clearly summarize the main results available in literature on the field. It is well established that the values of haematological and biochemical parameters can be affected by different physiological conditions as well as pathological circumstances. Therefore, the investigation of blood indices which may reflect responses to changes in homeostasis or disease is of paramount importance to identify animals with abnormalities. I think that the subject of the work is of interest and that the topic of the manuscript is appropriate for the Journal. The information is of significant interest to the Journal's readers. Some minor modifications are needed in order to improve the manuscript.
Specific comments:
The title is nice and accurately reflects the major findings of the work. Keywords represent the article adequately.
The summary as well abstract adequately summarize the rationale and the significance of the study.
The introduction section falls within the topic of the study. I suggest to add some references after the first sentence of introduction, please read and think to add “Guerrera M.C. et al., Marine and Freshwater Behaviour and Physiology, 2016, 49: 347-354; Arfuso F. et al., Theriogenology, 2017, 88: 145-151.”
Conclusion section is clear and well written. Indeed, Authors well summarize the findings available in literature on the field. However, I suggest to better emphasize the significance of the study. On this regards, please rewrite the last sentence of the conclusion section “In the conclusion of this review, many effects of bio-floc on fish aquaculture have been confirmed, and it can be used as a major standard for future mass production application in fish aquaculture.”
The tables are generally good and well represent the study. Whether possible, I suggest to add some figures in order to make more attractive the review for the reader.
Author Response
Reviewers' comments:
Reviewer 2.
1. The introduction section falls within the topic of the study. I suggest to add some references after the first sentence of introduction, please read and think to add “Guerrera M.C. et al., Marine and Freshwater Behaviour and Physiology, 2016, 49: 347-354; Arfuso F. et al., Theriogenology, 2017, 88: 145-151.”
Thank you for this suggestion. We read the article and added the reference "Guerrera M.C. et al., Marine and Freshwater Behaviour and Physiology, 2016, 49: 347-354; Arfuso F. et al., Theriogenology, 2017, 88: 145-151."
“In recent years, the rapid growth of the fish farming industry has been caused by many efforts to build intensive aquaculture farms and improve productivity [1,2]”
2. Conclusion section is clear and well written. Indeed, Authors well summarize the findings available in literature on the field. However, I suggest to better emphasize the significance of the study. On this regards, please rewrite the last sentence of the conclusion section “In the conclusion of this review, many effects of bio-floc on fish aquaculture have been confirmed, and it can be used as a major standard for future mass production application in fish aquaculture.”
Thank you for this suggestion. We added the sentence as below.
“In the conclusion of this review, various physiological effects and disease resistance of bio-floc on fish aquaculture have been confirmed. It provides important information to identify fish health which is directly related to fish production, and can be used as a standard for applying bio-floc to fish aquaculture in the future.”
3. The tables are generally good and well represent the study. Whether possible, I suggest to add some figures in order to make more attractive the review for the reader.
Thank you for this suggestion. We agree with this comment. Therefore, we added the graphic abstract
English editing is in progress, and it will be reflected in the final submission.
We sincerely thanks for your careful review to improve this article.

Reviewer 3 Report
The paper of You et al. entitled ˝Bio-floc technology in fish aquaculture: A review˝ tries to highlight the advantages of Bio-floc system in fish aquaculture. But the authors did not describe the system and presented only the experiments made with different fish species.
Several statements of authors are completely wrong from biochemical point of view:
Alpha, beta and gamma globulins are fractions of globulins and not subunits
Albumin is not involved in immune response
Triglycerides are not lipoproteins, but they are components of lipoproteins
Cortisol is not produced in kidney and liver
An increased MDA level is associated with a decreased defense system and not an increased one!
There are also drafting mistakes.
Unfortunatel
Author Response
Reviewers' comments:
Reviewer 3.
1. The paper of You et al. entitled ˝Bio-floc technology in fish aquaculture: A review˝ tries to highlight the advantages of Bio-floc system in fish aquaculture. But the authors did not describe the system and presented only the experiments made with different fish species.
Thank you for this suggestion. You have raised an important point here. However, our study is an integrated review focusing on the physiological changes of fish riased in bio-floc system rather than a description of the biofloc technology system. In the future, we will conduct research that strengthens the contents of the bio-floc system by applying the reviewer's opinion.
2. Several statements of authors are completely wrong from biochemical point of view:
Alpha, beta and gamma globulins are fractions of globulins and not subunits
Thank you for pointing this out. We agree with this comment. Therefore, we have revised the sentence as below.
Original sentence
“Globulin made up of subunits of a1, a2, β and γ-globulin is a critical component for maintaining a healthy immune system in fish, and an increase in globulin levels is associated with a stronger innate immune response [51].”
Revised sentence
“Globulin comprise a1, a2, β and γ-globulin fractions and is a critical component for maintaining a healthy immune system in fish. It is an increase in globulin levels is associated with a stronger innate immune response [55].”
3. Albumin is not involved in immune response
Thank you for this suggestion. You have raised an important point here. However, we believe that albumin would be more appropriate for immune responses according to many references.
We added references that albumin is related to the immune responses as follow:
Serum proteins, albumin, and globulin play a significant role in immune response, increases in serum protein, albumin and globulin levels are thought to be associated with a stronger innate immune response in fish [1].
Albumin and globulins are important serum proteins that play a critical role in immune response [2].
Serum proteins, albumin, and globulin play a significant role in immune response [3].
Increases in serum protein, albumin and globulin levels are thought to be associated with a stronger innate immune response in fish [3].
The increase in the levels of serum protein, albumin and globulin in fish is thought to be associated with a stronger innate immunity response [4].
Rises in total protein, albumin, and globulin are indicative of enhanced immunity responses in fish [5].
1. Yang, X., Guo, J.L., Ye, J.Y., Zhang, Y.X., Wang, W. The effects of Ficus carica polysaccharide on immune response and expression of some immune-related genes in grass carp, Ctenopharyngodon idella. Fish shellfish immunol. 2015. 42, 132-137. https://doi.org/10.1016/j.fsi.2014.10.037
2. Nandi, A., Banerjee, G., Dan, S.K., Ghosh, K., Ray, A.K. Probiotic efficiency of Bacillus sp. in Labeo rohita challenged by Aeromonas hydrophila: assessment of stress profile, haemato‐biochemical parameters and immune responses. Aquac. Res. 2017, 48, 4334-4345. https://doi.org/10.1111/are.13255
3. Chi, C., Jiang, B., Yu, X. B., Liu, T.Q., Xia, L., Wang, G.X. Effects of three strains of intestinal autochthonous bacteria and their extracellular products on the immune response and disease resistance of common carp, Cyprinus carpio. Fish shellfish immunol. 2014, 36, 9-18. https://doi.org/10.1016/j.fsi.2013.10.003
4. Akrami, R., Gharaei, A., Mansour, M.R., Galeshi, A. Effects of dietary onion (Allium cepa) powder on growth, innate immune response and hemato–biochemical parameters of beluga (Huso huso Linnaeus, 1754) juvenile. Fish shellfish immunol. 2015, 45, 828-834. https://doi.org/10.1016/j.fsi.2015.06.005
5. Mehrabi, Z., Firouzbakhsh, F., Rahimi-Mianji, G., Paknejad, H. Immunostimulatory effect of Aloe vera (Aloe barbadensis) on non-specific immune response, immune gene expression, and experimental challenge with Saprolegnia parasitica in rainbow trout (Oncorhynchus mykiss). Aquaculture. 2019, 503, 330-338. https://doi.org/10.1016/j.aquaculture.2019.01.025
4. Triglycerides are not lipoproteins, but they are components of lipoproteins
Thank you for pointing this out. We agree with this and have revised the sentence as below.
“Triglycerides, a provider of cellular energy are major components of lipoproteins along with cholesterol and phospholipids in fish, and it can be a critical biomarker to evaluate the nutritional status of fish metabolism [56].”
5. Cortisol is not produced in kidney and liver
Thank you for pointing this out. We agree with this comment. Therefore, we have modified the sentence as below.
“In fish under stress, cortisol secretion from interrenal cells in the head kidney is activated by corticotropin-releasing factors via the secretion of adrenocorticotropic hormone from the anterior pituitary, and the increase in cortisol is a major indicator to evaluate the stress in fish [63].”
6. An increased MDA level is associated with a decreased defense system and not an increased one!
Thank you for pointing this out. We agree with this comment. Therefore, we have revised the sentence as below.
“Liu et al. [72,77] reported a significant decrease in the MDA of O. niloticus cultured with bio-floc, which indicates that fish raised in bio-floc had adequate defense against lipid peroxidation and had better antioxidant capacity and regulatory mechanisms.”
“In addition, it was found that the MDA of C. carpio cultured with bio-floc was significantly decreased, which was due to improved the fish health and reducing oxidative stress [49].”
English editing is in progress, and it will be reflected in the final submission.
We sincerely thanks for your careful review to improve this article.

Reviewer 4 Report
I recommend minor revision. My comments and suggestion are present in the text. To see all please open the file in Acrobat Reader.

Author Response
Reviewers' comments:
Reviewer 4.
Thank you for this suggestion. We have revised the manuscript according to reviewer's suggestion.
English editing is in progress, and it will be reflected in the final submission.
We sincerely thanks for your careful review to improve this article.

Round 2
Reviewer 3 Report
The manuscript is improved after revision, but I still consider that the authors have to present briefly the bio-floc system. Despite the references cited, I have doughs regarding the involvement of serum albumin regarding the improvement of fish immune system. Can the authors to explain the biochemical or cellular mechanism by which albumin is involved in immune system.
I still consider that the authors have serious lack of knowledge in biochemistry of fish.
Author Response
Reviewer 3.
Reviewers' comments:
- The manuscript is improved after revision, but I still consider that the authors have to present briefly the bio-floc system.
: As the review's suggestion, we have added the sentence about the bio-floc system as follows.
“Bio-floc technology (BFT) is a more environmentally sustainable technology that uses beneficial microorganisms to absorb the ammonia and nitrite produced by feed waste, feces, and urine, which are naturally generated in the metabolic process of aquatic products [9]. This facilitates a self-nitrification process in aquaculture systems, without the exchange of stock water, and is achieved by stimulating the growth of beneficial microorganisms that can then be utilized as a feed source for aquaculture species and can absorb nitrogenous waste [10].”
“Innovative aquaculture system using BFT has been applied to many fish farms due to increasing concern about environmental pollution. BFT system is an eco-friendly closed system, which has many advantages including no water exchange, improved water quality and fish production, and less contamination by external factors [11]. BFT system requires the strong aeration and carbon sources such as sucrose, glucose, and molasses, and it helps to maintain the water quality by improving the activity of microorganisms and the removal of ammonia [5].”
- Despite the references cited, I have doughs regarding the involvement of serum albumin regarding the improvement of fish immune system. Can the authors to explain the biochemical or cellular mechanism by which albumin is involved in immune system.
I still consider that the authors have serious lack of knowledge in biochemistry of fish.
: We appreciate reviewer for pointing out the lack of explanation about albumin. As the review's suggestion, we have revised the sentence as follows.
Original sentence
Albumin present in serum proteins is a protein transporter synthesized by the liver and globulin and plays an important role in the immune system. Albumin and globulin are key proteins that have an important influence on the immune response, and an increase in albumin and globulin levels indicates a healthy immune status in fish.
Revised sentence
“Albumin and globulin are major protein in the serum; Albumin is a protein carrier involved in the transport of various substances including lipids, hormones, and Inorganic Ions [55]. Globulin comprises a1, a2, β, and γ-globulin fractions and is a critical component for maintaining a healthy immune system in fish, as an increase in globulin levels is associated with a stronger innate immune response [56]. The ratio of albumin to globulin is a useful indicator for monitoring fish health and immune status.”
English editing has been completed in all parts.
We sincerely thanks for your careful review to improve this article.
Round 3
Reviewer 3 Report
In this form, the paper can be accepted for publishing in Antioxidants journal.